# Respiration as a dynamic modulator of sensory sampling

Nikos Chalas[1,2], Martina Saltafossi[1,2], Teresa Berther[1,2], Elio Balestrieri[1,2], Omid Abbasi[1,2] & Daniel S. Kluger ®[1,2] ✉

Respiration dynamically modulates sensory perception by orchestrating transient states of the brain and the body. Using simultaneous recordings of high-density magneto- encephalography (MEG), respiration, and pupillometry, we show that human perceptual sensitivity to near-threshold visual stimuli was enhanced during inspiration, coinciding with respiration-modulated increases in arousal neuromodulation and cortical excitability. Participants adapted their breathing patterns to align with predictable stimulus onset, and this adaptive respiratory alignment correlated with improved performance. We further reveal that respiration-modulated changes in alpha and beta oscillations reflect distinct shifts in sensory and motor excitability, respectively. Crucially, respiration-resolved multivariate Granger causality analyses demonstrate that the breathing rhythm systematically shapes directed information flow within a widespread interoceptive network. This respiration-brain coupling was flexibly adjusted based on stimulus predictability, highlighting a mechanism for active sensing which integrates internal bodily rhythms with external sensory demands to optimise perception.

Recent work continues to highlight the link between bodily and neural rhythms and its role in human behaviour[1]. At rest, for example, the breathing rhythm is systematically coupled to so-called respiration-modulated brain oscillations (RMBOs) across a widespread anatomical network[2,3]. Consequently, there is now a substantial body of evidence showcasing respiratory modulation of human perception, particularly in the visual domain[4–6]. From an interoceptive inference perspective[7], the relationship between the breathing rhythm and human perception is by no means accidental: As respiration is uniquely under voluntary control, it may be used to actively align sampling of sensory information with neural and bodily states which facilitate performance. In humans, this application of active sensing[8] is hypothesised to implicate two processes in particular in order to entrain brain, body, and behaviour.

First, transient states of arousal neuromodulation describe a global physiological preparedness to process sensory information[9] and are therefore closely linked to human perception and cognition[10].

Under the control of the locus coeruleus (LC) and the nucleus basalis of Meynert, respectively, norepinephrine (NE) and acetylcholine (ACh) are released to widespread cortical regions[11] to regulate transitions between, e.g., attentional and behavioural states governing perception[12]. As NE release also affects pupil size, pupillometry has been established as an effective readout of arousal[13]. Not only have fluctuations in pupil-linked arousal been shown to influence cortical activity[14] and cognitive function[15], but recent work has rekindled the field's interest in respiratory modulations of arousal dynamics: Synaptic connections within the brain stem link the pre-Bötzinger complex - the central pattern generator of the respiration rhythm - to LC, suggesting that respiratory dynamics may influence arousal states[16]. Building on seminal rodent work demonstrating that interrupting the connection between both cores causes chronic hypoarousal and lethargic behavior in animals[17], it is now becoming increasingly clear that human respiration and pupil-linked arousal are systematically coupled[18]. Functionally, this link has predominantly

[1]Institute for Biomagnetism and Biosignal Analysis, University of Münster, Münster, Germany. [2]Otto Creutzfeldt Center for Cognitive and Behavioral Neuroscience, University of Münster, Münster, Germany. ✉e-mail: daniel.kluger@uni-muenster.de

been interpreted as serving the synchronisation of attentional states during perception, particularly in the visual domain[19].

A second vital determinant of perceptual processing is cortical excitability, i.e., the continuous balance between excitatory and inhibitory currents in the brain. In general, endogenous shifts in excitability states are a core characteristic of the brain as a dynamic system. Excitation-inhibition (E:I) balance within primary sensory cortices immediately affects behaviour, as e.g. excitability of the visual system determines whether a near-threshold stimulus would be perceived or not. The respiratory rhythm has been demonstrated as a modulator of both aperiodic[20,21] as well as oscillatory signatures of transient excitability states[2,4,22]. In behavioural paradigms, there is extensive evidence for an intricate connection between neural oscillations, excitability states, and behavioural performance: During visual perception, excitatory input to the visual cortex is regulated by functional inhibition in a feed-forward mechanism based on alpha oscillations (7-13 Hz), effectively controlling the excitability of the neural system per se[23]. This leads to an inverse relationship between prestimulus alpha power and excitability, corroborated by cross-modal evidence of strong ongoing alpha oscillations coinciding with reduced single-unit firing rates[24] and local field potentials[25]. In this context, and in line with the active sampling framework outlined above, the proposed role for the respiratory rhythm concerns the temporal alignment of information sampling with states of particularly high excitability[4]. In the motor domain - i.e., when responding to rather than perceiving sensory stimuli - respiration has similarly been linked to beta oscillations (15–40 Hz[5,26]), a prominent signature of predictive timing during perceptual processing[27].

To summarise, previous work has put the respiratory rhythm at the forefront of potential physiological modulators of behaviour. At present, converging evidence from different lines of research strongly suggests respiratory involvement in active sensing, including adaptation of respiratory behaviour to task timing[28], modulation of excitability and arousal states[4], and behavioural facilitation[29,30]. In the animal literature, active sensing has been suggested to comprise both a homeoactive and an alloactive component[31]. In this framework, alloactive sensing posits the use of mechanical energy to alter the parameters of the sensory apparatus, including voluntary changes in the respiratory rhythm driven by internal forward models of sensory predictions. What has so far been critically missing, particularly in humans, is a direct test of previous correlational findings within a dedicated predictive processing paradigm to answer fundamental open research questions: To what extent does respiration orchestrate excitability and arousal states across different contexts of stimulus predictability? What are the behavioural benefits of this respiration-brain coupling? And finally, how is the facilitative coupling of brain and body mechanistically achieved on the neural network level?

In the present study, we addressed these questions using simultaneous high-resolution magnetoencephalography (MEG) and respiratory data recorded during a cued visual detection paradigm. Specifically, we presented lateralised, near-threshold visual stimuli whose predictability was systematically manipulated with temporal and spatial cues. We report increased perceptual sensitivity for stimuli presented during the inspiration phase, coinciding with increased cortical excitability. This behavioural facilitation was achieved through adaptive adjustment of individual breathing patterns to task demands, with stronger adjustments linked to better performance. Pupil-linked arousal states depended on stimulus predictability and respiration phase, with a larger pupil diameter during predictable trials associated with improved performance during inspiration. Both alpha-band activity (indicating excitability states during visual perception) and beta-band activity (indicating predictive timing during motor preparation) were modulated by respiration phase and stimulus predictability. Multivariate source-level and Granger causality analyses further showed that respiration phase systematically orchestrated information flow within the RMBO network, directly influencing perceptual sensitivity. Collectively, these results underscore respiration as a highly adaptive modulator of neural dynamics and provide evidence for the neural mechanisms by which it facilitates behaviour.

## Results
### Behavioural performance

Participants performed a simple near-threshold visual detection task in which Gabor patches were briefly presented either left or right of a central fixation point. Critically, stimuli could be preceded by spatial and/or temporal cues which systematically manipulated stimulus predictability: During fixation, the neutral fixation cross could either be replaced by an arrow head (cued trials, C) and/or be surrounded by a circular countdown or not (timed trials, T; Fig. 1a). These manipulations allowed participants to predict the location and/or the temporal onset of the target stimulus. Upon button press, participants reported whether they had seen the target on the left, the right, or none at all. An adaptive QUEST staircase was used to adjust target contrast in a way that performance would settle at an overall hit rate (HR) of around $\mu_{HR} = 0.60$. On the group level, we observed an empirical mean hit rate of $HR_{total} = 0.55 \pm 00.02$ (M ± SD) across conditions.

The detection of near-threshold stimuli was facilitated when cue information was available during the fixation period. A linear mixed effect model (LMEM) revealed significantly higher hit rates for spatially cued ($HR_{cued} = 0.57 \pm 00.05$; M ± SD) vs uncued trials ($HR_{uncued} = 0.52 \pm 00.05$; $t(116) = 4.28$, $p < 0.001$) and for timed ($HR_{timed} = 0.56 \pm 00.05$) vs untimed trials ($HR_{untimed} = 0.54 \pm 00.06$; $t(116) = 20.00$, $p = 0.047$). The interaction term cueing X timing remained insignificant ($t(116) = -0.12$, $p = 0.901$); individual hit rates for each condition are shown in Fig. 1b.

### Respiratory phase information explains behavioural variance

Aiming first to replicate reports of general respiratory involvement in perception, we tested whether single-trial performance could be predicted from the respiratory phase angle at which a given stimulus was presented. As in a previous near-threshold perception study[4], we first set up a mutual information model to predict stimulus detection (hit, miss) from a combination of single-trial stimulus contrast, spatial cues (yes/no), and temporal cues (yes/no). Confirming the performance effects shown in Fig. 1b, including spatial and temporal cues significantly improved the binary prediction of correct vs. incorrect responses compared to a base model that only used single-trial contrast information ($t(29) = 2.64$, $p = 00.013$). In a second step, we also included sine and cosine of the respiratory phase at stimulus onset, which once again significantly improved the prediction of individual performance ($t(29) = 7.94$, $p < 00.001$). In sum, these results demonstrate that both stimulus predictability (spatial, temporal) and respiration phase - fully described by its sine and cosine components - had a significant effect on behavioural performance.

In light of these findings, we next aimed to characterise which periods of the respiratory cycle were driving these effects, i.e. at which respiratory phase behavioural performance was particularly modulated. Based on previous findings[4,6], we hypothesised a performance increase during the inspiratory phase. We exploited single-trial information regarding stimulus contrast and detection performance to iteratively refit the psychometric function. In short, the psychometric function expresses the sigmoidal relationship between the intensity of a stimulus and the probability of a particular response. In our analysis, we first fitted the overall psychometric function to all target trials (irrespective of condition), quantifying detection probability as a function of stimulus contrast. The resulting parameters were used as priors for a moving window approach in which we iteratively refitted the psychometric function to a subset of trials presented at a certain range of respiration phase angles. To this end, respiratory phase at the onset of each target stimulus was extracted and assigned to one of

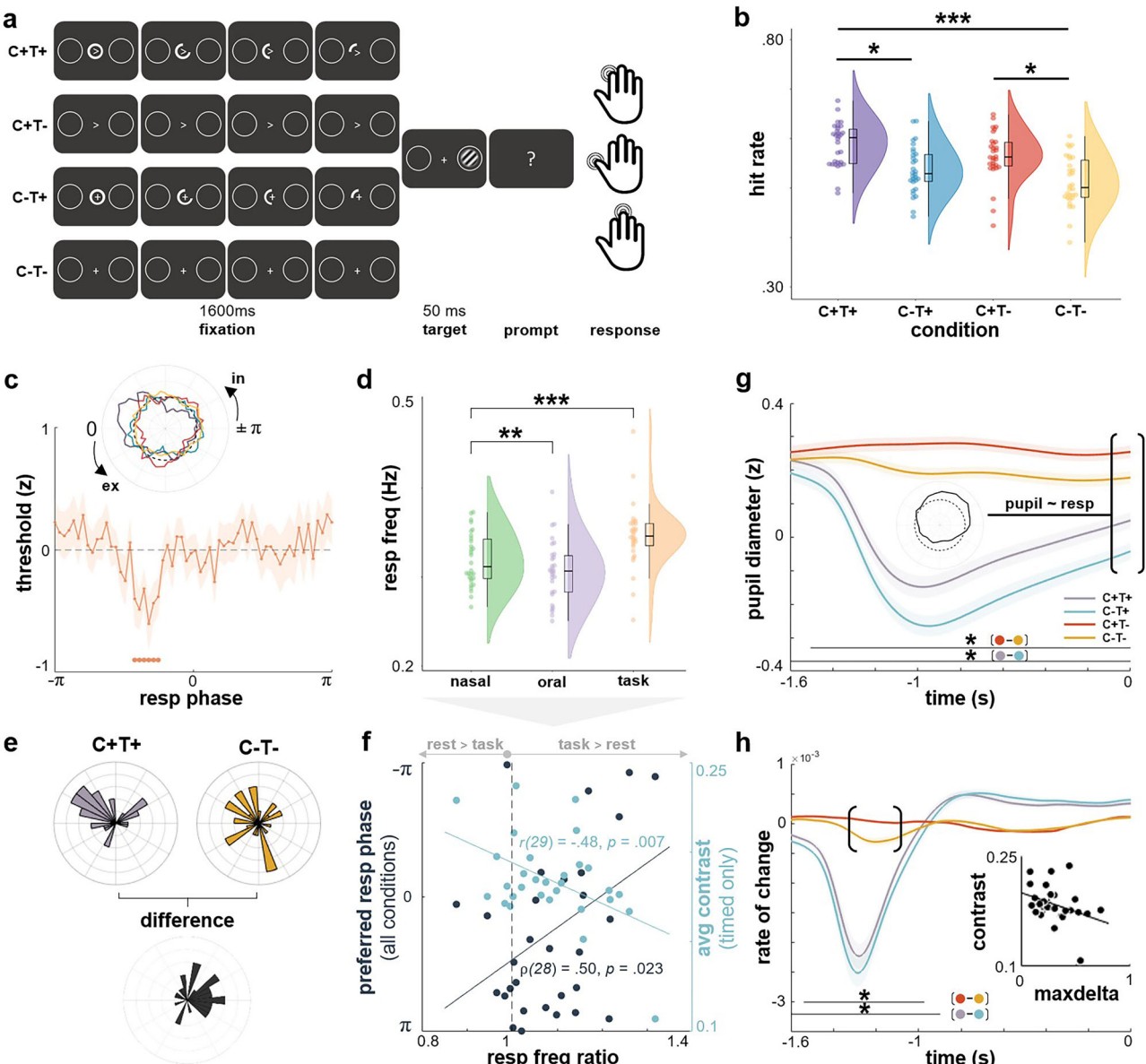

**Fig. 1 | Respiration modulates perceptual sensitivity and arousal states.**
**a** Participants detected visual stimuli presented on either side of the screen while fixating the centre. Stimulus detection was facilitated by spatial (*C*) and/or temporal (*T*) cues. **b** Pairwise comparisons revealed higher hit rates for *C + T +* (*p* < 0.001) and *C + T-* (*p* = 0.01) compared to *C-T-* (*n* = 180 trials per condition, two-sided, Tukey-Cramer-corrected). We observed higher hit rates for *C + T+* compared to *C + T-* (*p* = 0.01). Raincloud plots show median ± interquartile range. **c** Lower threshold for trials presented during inspiration (FDR-corrected, shown as M ± SEM) driven by maximally predictable targets (*C + T +*, see inset). The dashed line signifies the group-level mean. In = inspiration, ex = expiration. **d** At rest, *n* = 30 participants were breathing faster through their nose than through their mouth (two-sided Wilcoxon rank sum test, *z* = 2.60, *p* = 0.009). Nasal breathing during the task was faster than at rest (*z* = 3.67, *p* < 0.001). Raincloud plots show median ± interquartile range. **e** Respiration was actively adjusted for highly predictable targets so that they would be perceived during inspiratory high-sensitivity periods. **f** The more strongly

participants adapted their respiratory frequency during the task (compared to rest), the better their performance for timed trials (light blue, two-sided Pearson correlation). Circular-linear correlation between rest-to-state changes in respiration rate and preferred phase across all conditions suggests respiratory adaptation in favour of behavioural facilitation (two-sided, blue). **g** Spatial cues elicited robust increases in pupil diameter. Pupil diameter at target onset was strongly correlated with respiration phase (see inset; dashed line shows group-level mean across phase). Spatially cued trials lead to larger pupil diameter during fixation (M ± SEM). Pupil diameter is given in robust z-scores due to normalization during preprocessing. Asterisk indicates *p* < 0.05 for cluster-corrected t-tests. **h** The more strongly individual pupil diameter changed (maxdelta) when a spatial cue was present (vs when it was not), the lower the required stimulus contrast (M ± SEM). Asterisk indicates *p* < 0.05 for cluster-corrected t-tests. Source data are provided as a Source Data file.

k = 60 phase bins with all other trials presented at a similar respiratory phase (see[4] and Methods for details). Refitting the psychometric function for the subsets of trials assigned to each of these overlapping phase angle bins, we thus obtained a threshold estimate characterising perceptual performance at that respiration phase: Since the QUEST response criterion was fixed at $\mu_{HR}$ = 0.60, a lower threshold indicates higher sensitivity (ie lower stimulus intensity was sufficient to yield the

same level of performance). In support of our hypothesis, we observed a significant lowering of the perceptual threshold around the midway point of inspiration (ranging from −76° to −46°, see Fig. 1c), indicating an inspiration-locked increase in perceptual sensitivity across all trials. A subsequent analysis of individual conditions revealed that the beneficial inspiratory effect was driven by spatially and temporally predictable targets (C + T + ; see insert in Fig. 1c). It should be noted that

phase-related changes in decision criterion could occur simultaneously with the sensitivity modulation, but their computation would require changes in the experimental paradigm (i.e. the introduction of catch trials).

## The respiratory rhythm is adapted to task dynamics

Finally, since respiration is under voluntary control, we aimed to close the loop on respiration-locked performance modulations and investigated whether respiratory dynamics themselves would actively be adjusted to the temporal regime of the task in order to facilitate performance. Given the inspiratory facilitation for stimulus detection (Fig. 1c), the first question was whether this 'preferred phase' could be observed as phase-locking of single-trial data. To allow participants to actively adjust their breathing rhythm to predictable stimuli, we designed the perceptual task in such a way that the prestimulus period was kept consistent across conditions and long enough to potentially adjust the slow respiratory rhythm.

One way for participants to actively achieve stimulus occurrence at the phase of highest sensitivity (i.e., mid- to late inspiration) would be to adjust their breathing pattern to the perceived task timing. Hence, potential respiratory changes during the task would hint at a beneficial adaptation of breathing to the temporal regime of the task.

To directly analyse the potential facilitative effects of respiratory adaptation, we assessed participants' changes in respiratory rate (RR) from independent resting state recordings (nasal and oral breathing) to the task recording and related these RR changes to task performance. Nasal breathing during the task was significantly faster than nasal breathing at rest ($z = 3.67$, $p < 0.001$; Fig. 1d). Confirming the initial behavioural results, we did in fact observe clustering of single trials during the inspiratory phase for highly predictable C + T+ trials (Hodges-Ajne test against uniformity: $p = 00.005$), but not for C-T- trials ($p = 0.560$). A direct within-subject comparison between median angles revealed that this difference in preferred phases is indeed meaningful, as the mean circular difference between the two conditions was significantly different from zero ($p < 0.001$; Fig. 1e). Moreover, across all conditions, the more strongly participants adapted their respiratory frequency during the task (compared to rest), the better their performance (i.e. the lower the contrast necessary to achieve the targeted hit rate; see Fig. 1f). Critically, circular-linear correlation analysis revealed a robust positive correlation between rest-to-task change in respiration rate and preferred phase at stimulus onset for timed-only stimuli (C-T + ; $\rho(28) = 0.50$, $p = 0.023$). This finding strongly suggests that respiratory dynamics were indeed adjusted in such a way that temporally predictable stimuli would be perceived at the preferred respiratory phase associated with the highest perceptual sensitivity (Fig. 1f).

Overall, behavioural analyses showed that perceptual sensitivity for near-threshold stimuli was greatest during mid- to late inspiration and that participants capitalised on stimulus predictability to align respiration to target onset. Here, participants who achieved the best temporal alignment between respiration and target onset gained the strongest perceptual benefit.

## Arousal states reflect stimulus predictability and are linked to behaviour

Following up on these respiration-modulated behavioural facilitation effects, our next aim was to understand their neural signatures. Given that pupillary dynamics provide a well-established proxy of noradrenaline-driven changes in arousal states[13], we investigated pupil responses during the prestimulus window, i.e., between cue onset and target onset. Controlling for lighting changes on the screen (i.e. the presence of an additional bright stimulus during C-T+ and C + T+ trials, see Fig. 1a), spatial cues significantly modulated arousal during prestimulus fixation leading up to target presentation, irrespective of temporal cues (FDR-corrected $p < 0.05$; Fig. 1g), most likely reflecting

the pre-stimulus difference in target uncertainty. Pupil diameter at target onset (across all conditions) was systematically related to respiration phase at target onset (Rayleigh test of individual circular means: $z(29) = 8.51$, $p < 0.001$), with greater diameter indicating higher arousal during inspiration. In addition to the reported overall effect, this modulation was consistently observed within each condition as well. Moreover, larger pupil diameter changes as a response to cued (vs uncued) trials coincided with lower average stimulus contrast on the individual level ($\rho(28) = -0.38$, $p = 0.019$), indicating a systematic influence of predictability- dependent changes in arousal states on behavioural performance (Fig. 1h). As a control analysis, we quantified saccadic behaviour during the prestimulus time window across the four conditions. As expected, the number of saccades overall was low (with the group-level median ranging from M = 4 to M = 6 saccades over 180 trials across conditions) and a Kruskall–Wallis test showed no group-level differences in the number of saccades across conditions ($p = 0.92$).

## Distinct oscillatory signatures of behaviourally relevant stimulus characteristics

Moving from arousal neuromodulation to excitability states, we next sought to confirm a well-established neural signature of perceptual performance, namely alpha suppression effects over parieto-occipital sensors for hits vs misses (i.e., perceived vs non-perceived trials; Fig. 2a). Replicating previous work from our lab[4], we report long-lasting alpha suppression for perceived trials (Fig. 2b), starting as early as the cue onset at -1600 ms prestimulus. Critically, these alpha suppression effects were related to changes in pupil-linked arousal in two ways: Pooling all conditions, we first observed a consistent increase in pupil size (indicating higher arousal) for hits vs misses (FDR-corrected $p < 0.05$; Fig. 2a, upper inset). Second, the greater this diameter increase for hits vs misses on the individual level, the stronger the alpha suppression we observed ($\rho(28) = -0.42$, $p = 0.011$; Fig. 2a, lower inset). Alpha suppression effects temporally preceded changes in pupil diameter (Fig. 2c).

Complementing our behavioural findings outlined above, we next characterised the effects of spatial and temporal cueing on the neural level, particularly changes in prestimulus alpha and beta power. We hypothesised distinct effects of spatial cues on perceptual processing and of temporal cues on predictive timing during motor preparation, respectively. A conjunction analysis of spatially cued vs uncued trials revealed an occipital distribution of significant alpha power decrease around −1400 ms to −600 ms prior to stimulus onset (Fig. 2d). Conversely, beta suppression effects for timed vs untimed stimuli were observed later (around −1000 ms up to stimulus onset) and localised over central sensors (Fig. 2e). Descriptive statistics for cluster peaks are provided in Fig. 2f.

## Respiration-driven spectral changes within the RMBO network mediate perception

Going beyond sensor-level analyses, we had previously identified the RMBO network as a set of nodes in which neural oscillations were systematically modulated by the respiratory rhythm[2]. Here, we capitalised on these anatomical priors to investigate if and how the effect of stimulus predictability on respiration and behavioural performance would be mediated through changes in the RMBO network. To this end, we conducted source-level analyses using an LCMV beamformer. Using the parcellation atlas by Glasser and colleagues[32], we extracted source-level time series for specific cortical regions of interest (ROIs) taken from the RMBO network. Each of these ROIs consisted of 2-6 parcels and comprised bilateral insula (INS), anterior and posterior cingulate cortex (ACC, PCC), supplementary motor area (SMA), and temporoparietal junction (TPJ). In addition to these nodes of the RMBO network, we included primary and secondary visual cortex (V1/2) as well as primary motor cortex (M1) as domain-specific upstream

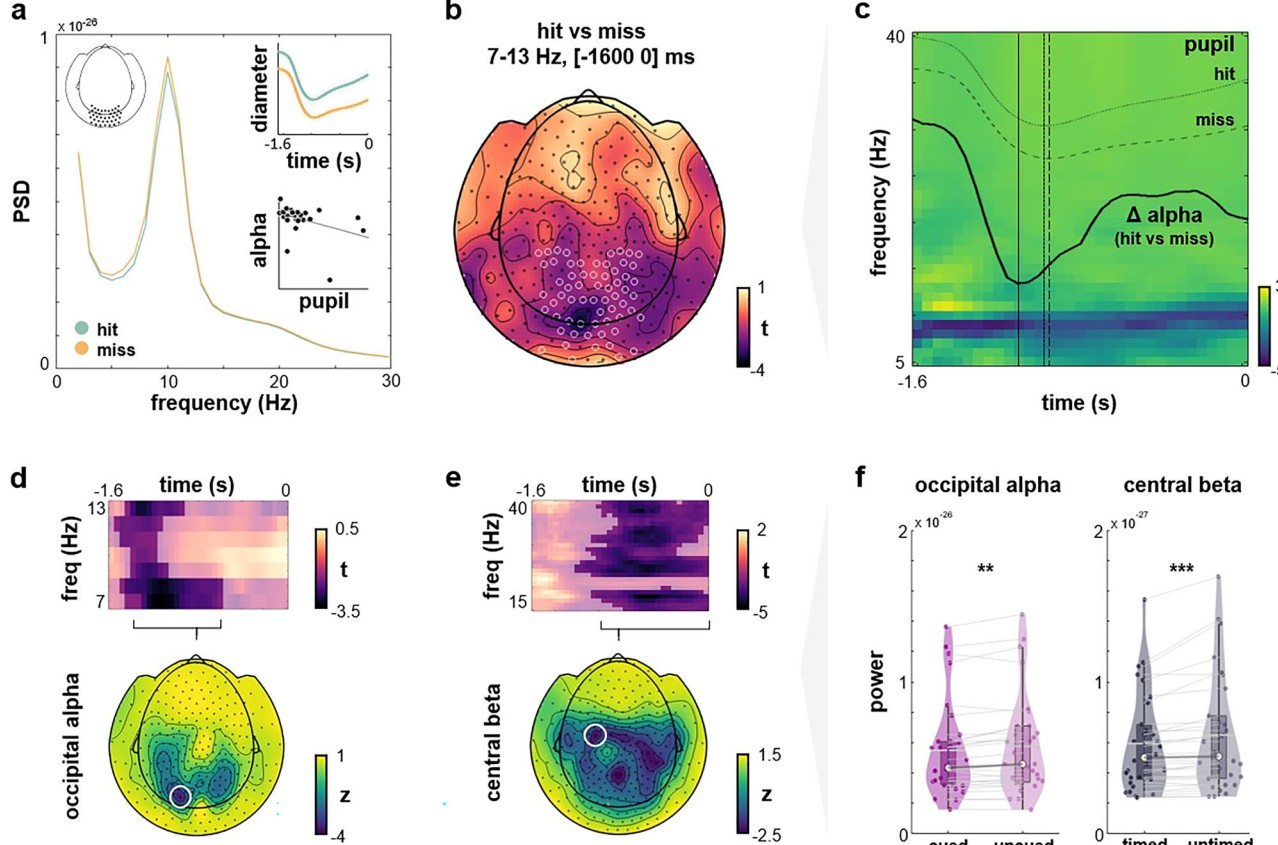

**Fig. 2 | Alpha and beta modulations reflect distinct but behaviourally relevant dimensions of stimulus predictability. a** Power spectra for hits vs misses with posterior alpha suppression. Top inset: Pupil diameter (arb.units) increased for hits vs misses across all conditions (M ± SEM). Bottom inset: Negative correlation between prestimulus alpha power and diameter. **b** Topography of cluster-corrected t-values for prestimulus alpha power extracted from hits vs misses. Markers indicate sensors with significant alpha band differences when cluster-correcting across time, frequencies, and sensors. **c** Time-frequency representation averaged across posterior sensors (marked in b) shows pre-stimulus alpha sup-pression for hit vs miss trials. The time course of this suppression effect is super-imposed in bold; pupil traces for hit and miss trials follow alpha modulation. **d** Top: Pre-stimulus alpha frequency modulation over visual cortices induced by spatial cueing ($[C+T-\vee C+T+]$ vs $[C-T-\vee C-T+]$; all trials mapped onto the right hemi-sphere). Bottom: Topography of significant alpha suppression (averaged between

[-1400 -600] ms). **e** Top: Prestimulus sensorimotor beta modulation for timed $[C+T+\vee C-T+]$ vs untimed $[C+T-\vee C-T-]$ trials. Bottom: Topography of significant beta suppression (averaged between [-1000 0] ms). **f** Individual alpha power for cued vs uncued targets (left) and beta power for timed vs untimed targets (right, $n=360$ trials per collapsed condition). Alpha power was extracted from an occipital sensor (marked in panel **d**) and averaged between 7-8 Hz and [-1200 -1000] ms (cluster peak from **d**). Beta power was extracted from a left-central sensor (marked in e) and averaged between 24-35 Hz and [-800 -400] ms (cluster peak from **e**). As to be expected from panels d and e, we observed lower alpha power for uncued vs cued targets (two-sided t-test, $t(29) = -3.32$, $p = 0.003$, $BF_{10} = 15.25$) and lower beta power for untimed vs timed targets ($t(29) = -3.72$, $p < 0.001$, $BF_{10} = 38.42$). Note that we provide these statistics solely for illustrative purposes, as the significance of the effect had already been established. Violin plots show median ± interquartile range. Source data are provided as a Source Data file.

nodes involved in visual processing and motor preparation, respec-tively (Fig. 3a). After correcting for head motion artefacts, we first computed prestimulus power spectra for each ROI (DPSS multitaper, frequency range 1-40 Hz, [-1600 0] ms) to quantify oscillatory activity leading up to target onset. In order to remove the aperiodic compo-nent from the frequency spectra of each trial, we considered the first-order derivative of each time series, which can be used as a time-domain filter of the aperiodic component[32]. Visual inspection of the resulting power spectra showed that this spectral whitening did in fact remove the 1/f dependency from our data (Fig. 3c; also see Supple-mentary Fig. S1 for replication of the results without spectral whiten-ing). Using a moving-window approach, we computed power spectra for 30 respiration-phase bins per ROI and per participant. Based on these individual matrices of ROI x frequency x phase bin, we per-formed a circular-linear correlation[33] of frequency-specific power with the corresponding phase angle. Across all trials, significant group-level correlations between oscillatory power and respiration phase were found for almost the entire frequency spectrum across ROIs (Fig. 3b;

FDR-corrected $p < 0.05$). These circular-linear correlations quantify systematic covariation of oscillatory power across the respiratory phase. This relationship was particularly evident in alpha and beta frequency bands (Fig. 3c, top panels), in that e.g. alpha power within the visual cortex was systematically lowered during inspiration (com-pared to expiratory phase bins). This link demonstrates the funda-mental modulatory influence of the respiratory rhythm on oscillatory neural network activity prior to the onset of near-threshold stimuli.

To better understand these baseline changes in oscillatory activity within the RMBO network, we next investigated target predictability as a potential modulating factor. To this end, we statistically compared prestimulus power spectra for maximally predictable $(C+T+)$ and unpredictable targets $(C-T-)$, irrespective of respiration phase at target onset and behavioural performance. We report a wide-range, con-sistent power decrease for predictable targets in almost all cortical parcels corresponding to our regions of interest (Fig. 3d, cluster-corrected $p < 0.05$). Although significant power modulations were observed across theta, alpha, and beta frequency bands, most ROIs

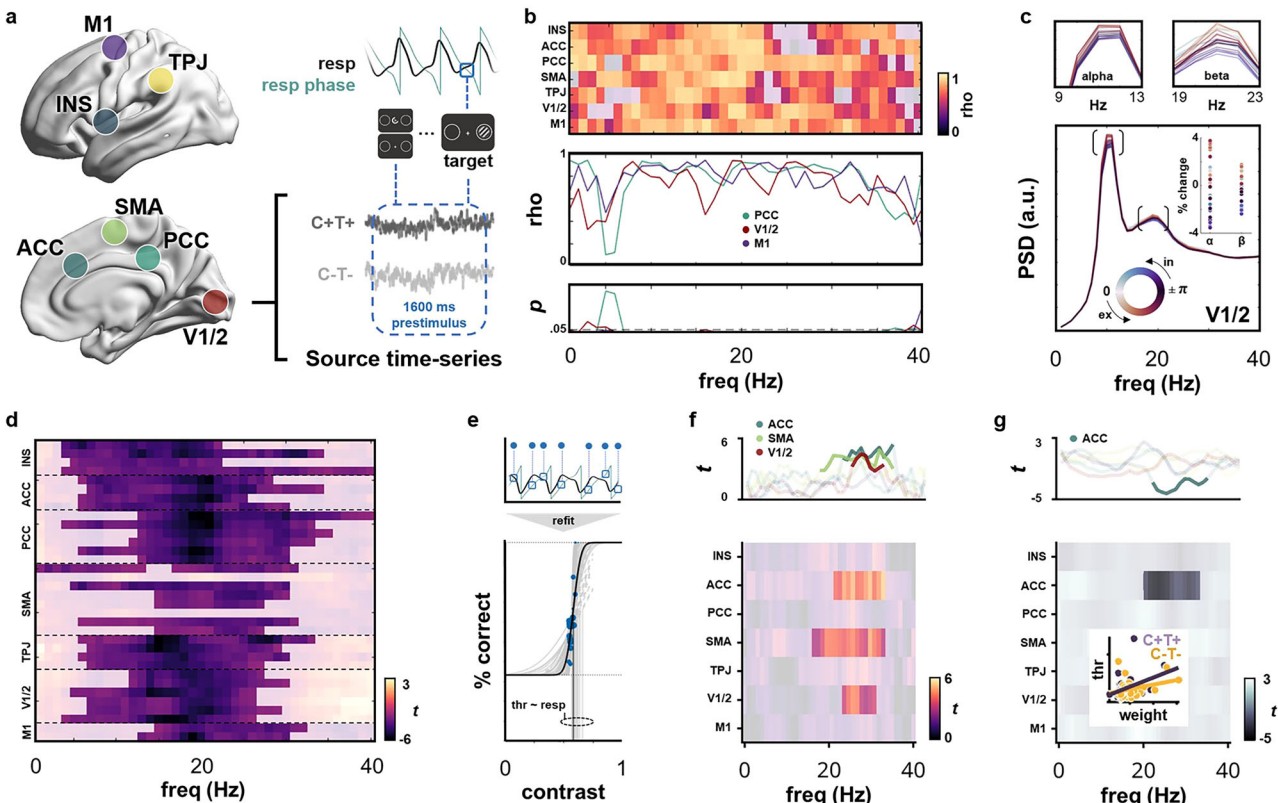

**Fig. 3 | Respiration governs sensory sampling through modulations of neural network activity. a** Source-level time series of prestimulus activity ([-1600 0] ms) extracted from RMBO network ROIs, including insula (INS), temporoparietal junction (TPJ), supplementary motor area (SMA), posterior cingulate cortex (PCC), and anterior cingulate cortex (ACC), as well as from visual (V1/2) and motor cortices (M1). The respiratory phase was extracted at the target onset. **b** Circular-linear correlation across respiration phase at target onset and single-trial power in individual frequencies (0–40 Hz) for each functional ROI, collapsed across experimental conditions. **c** Exemplary respiration phase-resolved power spectra extracted from V1/2, collapsed across experimental conditions. The inset illustrates the percentage change in power for peaks in the alpha (9–13Hz) and beta (19–23Hz) bands. The top panels show alpha and beta peak frequency ± 2 Hz, respectively. **d** Cluster-based, FDR-corrected permutation statistics on power spectra for maximally unpredictable vs predictable targets ($C$-$T$- vs $C$+$T$+), irrespective of respiration phase. **e** Single-participant example of refitting the psychometric function across the whole data set (bold black line, also see Fig. 1c). Threshold variation across respiration phase bins is shown in grey. **f** We conducted a subject-level regression between respiration phase (i.e., sine and cosine) and frequency-specific power (0–40 Hz) on all trials (all conditions). This resulted in regression coefficient spectra, i.e., one regression weight per frequency, ROI, and condition. Regression weights were correlated with individual threshold estimates from the psychometric function (**e**). We show ROI-wise statistics for the group-level correlation between phase-locked power changes and perceptual sensitivity. Opacity mask indicates a significant relationship between respiration, beta-band oscillations, and perceptual sensitivity within ACC, SMA, and V1/2 (cluster-corrected). **g** Same analysis as in (**f**), but now for a subject-level regression performed separately for maximally predictable ($C$+$T$+) and unpredictable targets ($C$-$T$-). We show group-level correlations between individual regression coefficient spectra differences ($C$-$T$- minus $C$+$T$+) and the respective differences in psychometric function thresholds. Group-level correlation of respiration-locked ACC beta-band oscillations and perceptual sensitivity differentiated between predictable and unpredictable trials (cluster-corrected). The inset shows stronger beta-band correlation for predictable vs unpredictable trials. Source data are provided as a Source Data file.

---

showed the strongest suppression effects in beta frequencies between 18-22 Hz.

In a final step, we aimed to relate respiration phase-locked changes in neural oscillations to individual differences in perceptual sensitivity. For this purpose, our approach was twofold: First, we sought to confirm the overall effect of the respiratory phase on prestimulus spectral power within our network of interest. To this end, we performed a linear regression per participant, frequency, and ROI, predicting frequency-specific power from respiratory phase (as a linear combination of sine and cosine) at target onset. This first-level analysis yielded regression coefficients per participant, frequency, and ROI. For the perceptual component, we refitted the psychometric function (relating performance to target contrast) for each participant across all trials, irrespective of respiratory phase, and extracted the function's threshold as a measure of perceptual sensitivity (see Figs. 1c and 3e). Linking respiratory modulations of brain oscillations (RMBOs[2]) to perception, we then correlated individual beta coefficient spectra from our regression analysis with individual thresholds extracted from the

psychometric function. On the group level, we observed clusters of significant correlations between RMBOs and perceptual sensitivity within beta-band frequencies located in ACC, SMA, and V1/2 (Fig. 3f; cluster-corrected $p$ <0.05). In other words, stronger respiratory modulation of beta activity in these ROIs coincided with greater differences in perceptual sensitivity. The directionality of this effect becomes evident when we consider the respiration phase-locked beta modulations shown in Fig. 3c: Prestimulus beta power was lowest (i.e., suppressed most strongly) when targets were presented during mid-to-late inspiration, which is precisely the respiratory phase for which we observed significantly increased perceptual sensitivity on the group level (Fig. 1c). Collectively, these findings demonstrate that respiration-related changes in neural oscillations immediately affect perception: Not only did we show suppressed alpha power over occipital sensors for hit vs miss trials (Fig. 2a-c), but then revealed a significant correlation between respiration phase and alpha power in V1/2 with lower alpha power during inspiration (compared to expiration, Fig. 3b, c). In a dedicated correlational analysis, we further established that

perceptual processing was mediated by the link between respiration and beta-band activity within medial central (ACC, SMA) and visual cortices (V1/2), relating respiration-locked beta suppression to increased perceptual sensitivity.

Finally, we asked whether this relationship between respiration-modulated neural oscillations and perceptual sensitivity would be altered by the predictability of upcoming stimuli, in that highly predictable stimuli allow for coordination of brain and body in favor of behavioural facilitation. Hence, we repeated our previous analysis separately for maximally predictable targets preceded by both a spatial and a temporal cue (C + T + ) and unpredictable targets which were not preceded by any cue (C-T-). We once again used linear regression to predict prestimulus spectral power from respiration phase at target onset and then correlated differences in the resulting regression coefficient spectra with the respective individual threshold differences (i.e., perceptual sensitivity for C + T+ minus C-T-). We found that the previously reported respiration-brain-behaviour correlation within ACC beta-band activity reliably distinguished between predictable and unpredictable targets (Fig. 3g; cluster-corrected $p$ <0.05). This interaction with target predictability demonstrates that respiratory involvement in perceptual processing was dynamically adjusted to momentary perceptual demands, as we observed a stronger respiration-brain-behavior relationship when cue information was available prior to target onset, allowing for respiratory adaptation. In sum, our results thus far indicate that the rhythmic act of breathing is systematically coupled with neural oscillations throughout the brain, particularly in the alpha and beta frequency bands. As these oscillations reflect excitation-inhibition dynamics within visual and motor cortices in our visual detection paradigm, respiration is voluntarily adapted to facilitate perceptual processing by temporally aligning sensory sampling with high-excitability states. This respiration-driven brain-body coordination is particularly effective when upcoming sensory information is highly predictable in both location and timing, underscoring the system's remarkable ability to flexibly adapt to short-term contextual changes. Naturally, our final aim then was to elucidate how this facilitative coupling of brain and body would be mechanistically achieved on the RMBO network level.

**Respiration shapes perception through changes in directed network connectivity**

From multivariate source-localised time series, we computed power and cross-spectral densities (DPSS multitaper, 0–150 Hz) for our set of RMBO nodes plus V1/2 and M1 (see Methods for details). Based on these frequency representations, pairwise multivariate Granger causality (mGC) was computed for all pairs of ROIs during the prestimulus period (Fig. 4a). By considering the combined influence of multivariate source-level time series, multivariate Granger causality allows for superior forecasts compared to conventional approaches. To quantify the extent to which respiration phase modulated directed functional connectivity within the RMBO network, we once again sorted all trials (collapsed across conditions) according to respiratory phase at target onset and then used a moving window approach to compute respiration phase-resolved mGC spectra (Fig. 4b). In keeping with our earlier approach (see above), we conducted a first-level linear regression predicting mGC per frequency and pairs of nodes from respiratory phase. We then computed the directed asymmetry index (DAI[34,35]) for the resulting regression weights as a measure of frequency-specific directionality between any two nodes (Fig. 4c). As positive DAI values indicate a stronger A → B connectivity and vice versa, we finally tested the DAI of respiration phase-resolved mGC spectra against zero using cluster-permutation correction. On the group level, this analysis revealed that respiration phase modulated broadband directed connectivity within the RMBO network, particularly involving insula, ACC/PCC, TPJ, and primary as well as supplementary motor areas (Fig. 4d). All pairwise DAI spectra are shown in Supplementary Fig. S2.

Exemplary grand-average spectra of GC regression weights for the connection between insular cortex and TPJ are shown in Fig. 4e along with the resulting asymmetry spectrum (Fig. 4f; see Supplementary Fig. S3 for all pairwise interactions). Similarly to the GC spectra, we observe variable peaks in the delta, alpha, and beta frequency bands, reflecting respiration-modulated interactions between INS and TPJ. With regard to the directionality of this connection, INS-TPJ connectivity was significantly modulated by respiration in frequencies up to 30 Hz (Fig. 4d), in that the dominance of INS → TPJ information flow was systematically higher during expiration (Fig. 4c, red traces) compared to inspiration (blue).

In addition to the overall influence of respiration on directed RMBO network connectivity - i.e., respiration-phase-dependent modulations shared across all experimental conditions - we were particularly interested in whether these connectivity changes would vary with target predictability and whether they were systematically related to perceptual processing. To this end, we repeated the mGC regression analysis separately for maximally predictable targets (C + T + ) and unpredictable targets (C-T-). On the group level, we found a significant influence of respiration phase on broadband connectivity for a wide range of pairwise connections within the RMBO network which were markedly similar for predictable and unpredictable targets (Fig. 4g; cluster-corrected $p$ < 0.05). In fact, the sole connection for which we observed a condition X respiration phase interaction on directed connectivity was the link between TPJ and visual cortices (V1/2). As shown in Fig. 4h, connectivity asymmetry between these two nodes in the alpha band (around 9-15 Hz) was differentially related to target predictability: When both temporal and spatial information about an upcoming target was available (C + T + ), we observed a stronger respiratory modulation of prestimulus alpha-band information flow from TPJ to V1/2, compared to trials for which no prior information was available (C-T-). In a final analysis, we therefore investigated whether the functional link between respiratory phase, RMBO network connectivity, and perceptual sensitivity depended on the amount of sensory information available. Based on individual differences in respiration-modulated DAI spectra (C + T+ vs C-T-), we conducted a correlation analysis with the respective condition difference in individual perceptual thresholds (see Figs. 1c and 3e). Using non-parametric cluster permutation statistics, we observed a significant circular-linear correlation between alpha-band connectivity in the INS-SMA connection and perceptual sensitivity during C + T+ trials (Fig. 4i; cluster-corrected $p$ <0.05). This correlation was strongest at around 8 Hz, at which point group-level DAI indicated balanced information flow between the insular and supplementary motor cortex. On the individual level, the difference between INS-SMA (a)symmetry for predictable (vs unpredictable) trials was negatively correlated with the difference in perceptual threshold between the two conditions (Fig. 4j). In other words, for predictable targets only, the increase in alpha-frequency SMA → INS connectivity prior to target presentation was associated with higher perceptual sensitivity in perceiving these targets (i.e., lower threshold; Fig. 4k). Collectively, our analyses of directed RMBO network connectivity revealed that respiration phase significantly modulated broadband connectivity across RMBO nodes, particularly connections involving insula, anterior and posterior cingulate, TPJ, and (supplementary) motor areas. Of these pairwise links, stronger respiratory modulation of low-frequency SMA → INS connectivity coincided with greater behavioural facilitation in perceiving predictable (vs unpredictable) targets, suggesting a specific functional pathway by which respiration adjusts perceptual processing to momentary contextual demands and facilitates behaviour.

## Discussion

We present comprehensive neurophysiological and behavioural evidence demonstrating how the respiratory rhythm governs the interplay of cortical arousal and excitability states for the benefit of

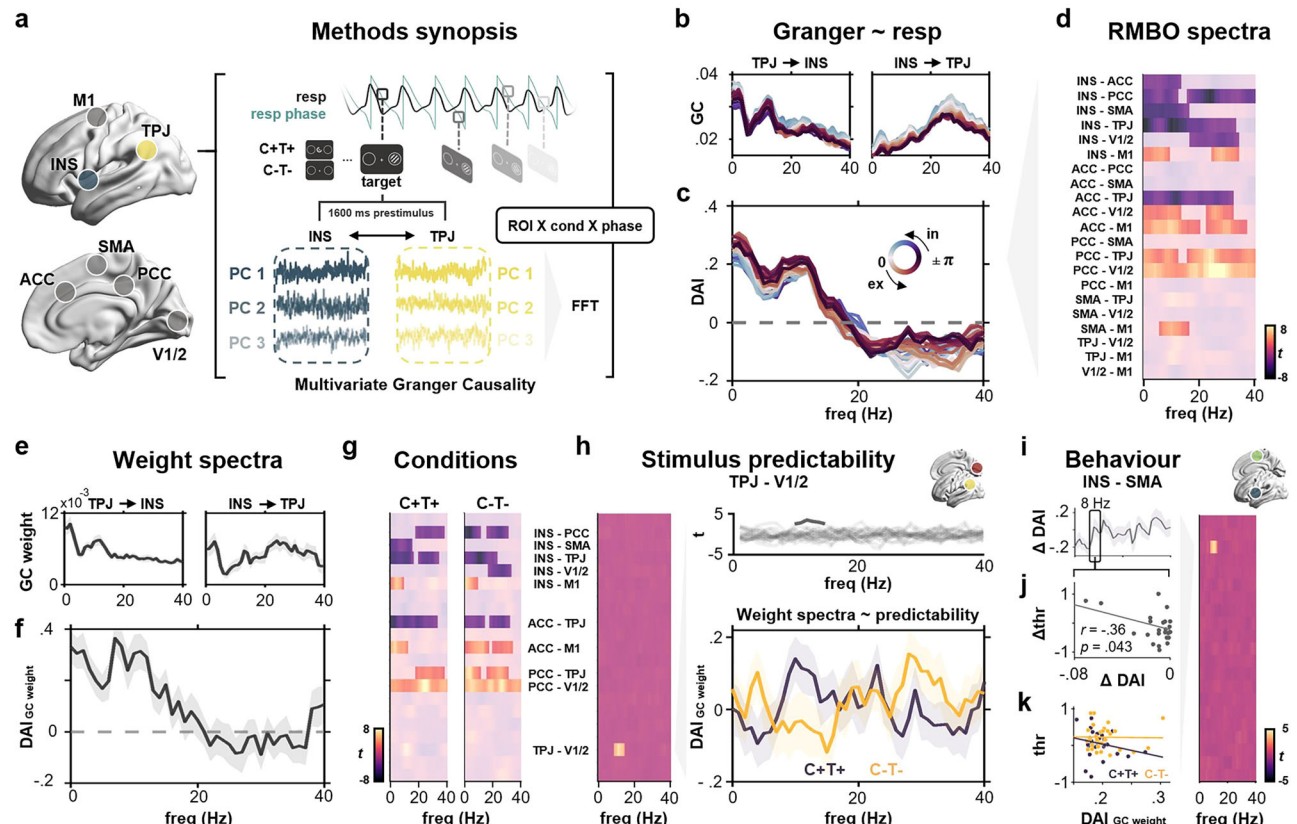

**Fig. 4 | Respiration-driven changes in directed RMBO network connectivity shape context-dependent perceptual processing. a** Multivariate source-level time series of prestimulus activity ([−1600 0] ms) were extracted for each condition from each ROI for pairwise multivariate Granger causality (GC) on frequency-resolved data. A moving window across respiratory phases at target onsets allowed us to estimate phase-resolved multivariate GC. **b** Exemplary grand-average GC spectra between insula and TPJ across the respiratory phase at the target onset. **c** Directed asymmetry index (DAI) of the GC spectra between insula and TPJ. Positive values indicate stronger TPJ-to-INS influence. Note the low-frequency DAI modulation during inspiration (blue) compared to expiration (red). In = inspiration, ex = expiration. **d** We performed first-level, frequency-specific regression of multivariate GC and respiration phase at target onset. Significant respiratory modulation of broadband directed functional connectivity was found for pairwise connections between insula, ACC/PCC, TPJ, and primary/supplementary motor cortices. **e** Exemplary grand-average regression weight spectra between insula and TPJ (shaded areas show SEM). **f** Regression weight spectra for INS → TPJ (see d) were converted to DAI spectra (shaded areas show SEM). **g** Cluster-corrected group-level statistics of regression weight spectra (see e) for all pairwise ROI interactions, extracted from RMBO network connectivity spectra prior to maximally predictable ($C+T+$, left) and unpredictable targets ($C$-$T$-, right). **h** Contrast of [$C+T+$ vs $C$-$T$-] trials (shown as M ± SEM) revealed a significant effect of stimulus predictability on alpha-band DAI between TPJ and V1/2, ranging from around 8 - 15 Hz. **i** For each participant, we computed frequency-wise correlations between the DAI interaction between the respiration phase and the experimental condition ($C+T+$ vs $C$-$T$-) and the perceptual threshold difference between the two conditions. Cluster-corrected statistics for this correlation revealed alpha-band modulation in the INS-SMA connection (M ± SEM). **j** Negative correlation between $C+T+$ vs $C$-$T$- differences in threshold and DAI from the respiratory regression on GC, respectively (two-sided Spearman correlation). **k** Condition-wise correlations between thresholds and DAI of the GC regression weight, extracted for alpha-band asymmetry between insula and SMA at 8 Hz (cluster peak from **h**). Source data are provided as a Source Data file.

perceptual processing. Participants showed greater perceptual sensitivity for near-threshold stimuli presented during inspiration and adjusted their respiratory pattern accordingly when spatial and temporal cueing information was available. Behavioural performance was dependent on transient changes in arousal neuromodulation and cortical excitability of the visuomotor system, which was in turn systematically modulated by the respiration phase. Source-level analyses of directed functional connectivity revealed respiration-modulated broadband connectivity changes in a network comprising visual and insular cortex, anterior and posterior cingulate, TPJ, and (supplementary) motor areas. Again, this respiratory coupling to neural dynamics was linked to perceptual changes and shown to be flexibly and voluntarily adjusted depending on momentary contextual information. In what follows, we ground our findings in theoretical accounts of predictive processing and set out the central implications for our understanding of how brain-body states are orchestrated.

More and more theoretical proposals suggest that respiratory predictive processing is well suited to explain the computational mechanisms by which the breathing rhythm modulates brain activity and behaviour[7,36]. Lending from active sensing accounts proposed in the animal literature[8], these accounts posit that multiple streams of sensory information - e.g., internal and external signals - are coordinated in a way that optimises their integration and propagation[36]. Since respiration is under voluntary control, it is a prime example of an internal signal by which the system can exert alloactive control over sensory inputs, e.g. by temporal alignment through dynamic changes in the pace of the respiratory rhythm we observed in the present study. This predictive timing has been shown to maximise the signal-to-noise balance between interoceptive and exteroceptive processing[37] and synchronise sensory sampling with favourable brain states of excitability[4,22] and arousal[18]. Despite this convincing line of primary evidence, a dedicated investigation of the functional mechanisms underlying respiratory predictive processing has so far been lacking. For the first time, the present study provided direct insight into the predictive nature of respiration-brain coupling. As we and others have previously suggested that this coupling is implemented through

systematic modulations of hierarchically organised neural oscillations[38,39], much can be gained from understanding the functional roles assigned to oscillations of different frequencies within predictive processing accounts of brain function.

Here, a particularly strong case can arguably be made for the role of alpha oscillations in visual perception, both with regard to states of excitability and arousal. Recent research has highlighted that pupil-linked arousal coincides with widespread changes in cortical activity, including modulation of alpha activity in the visual cortex[14], closely tied to perceptual sensitivity near threshold[40]. As for brain-body involvement, the respiratory cycle has been shown to modulate perceptual decision thresholds, particularly during inhalation[4,6]. Note, however, that some studies report expiratory facilitation of behaviour[6,41], suggesting that the distinction between inspiration- and expiration-locked modulation may depend on whether the events of interest primarily concern sensory perception or an immediate behavioural response. In the present study, respiration-modulated changes in alpha activity temporally preceded shifts in pupil dilation, indicating that arousal neuromodulation is primed in neural circuits and, as a result, affects perceptual processing[4]. Fittingly, spatial cues - predictive information about where to expect an upcoming stimulus - induced a decrease in alpha power and larger pupil dilation leading up to stimulus presentation. These neural and pupillary indicators of increased excitability and arousal are in line with previous work showing that expectation and attentional engagement can modulate both neural oscillatory activity and arousal markers[42]. Particularly, larger pupil size in the context of predictive cues has been linked to reduced uncertainty and enhanced perceptual precision[43], while alpha suppression is associated with enhanced attentional allocation to the anticipated stimulus[44] – both consistent with the notion that anticipatory attention and arousal jointly optimise perceptual decision thresholds. These findings corroborate the proposal of human active sensing by demonstrating that both endogenous respiratory control and exogenous information are jointly converged to coordinate transient states of upregulated cortical excitability (indexed by alpha power) and arousal (indexed by pupil dilation) in service of perceptual sensitivity.

Critically, present respiration-related changes in oscillatory markers of excitability states were by no means restricted to the alpha band or the visual system: Within the RMBO network, we further found prestimulus beta-band suppression to be modulated by respiratory phase. Decreases in beta power were correlated with increased behavioural performance irrespective of the predictability of the stimulus. This aligns with previous accounts of beta power as a reflection of inhibitory states impairing stimulus detection in the visual[45] and auditory modalities[46]. Beta rhythms are generally associated with the preparation and control of action as well as the initiation of motion onset[47], preparing the system for action prior to perception. Specifically, a recent study conclusively demonstrated that sensorimotor excitability states reflect beta-band oscillatory activity[48]. Our findings extend this framework and strongly indicate that the respiratory rhythm poses a fundamental biological constraint on motor system excitability and action preparation in perception. This interpretation of predictive timing was substantially corroborated by three key findings: First, temporal cues - predictive information about when to expect an upcoming stimulus - elicited strong beta suppression effects over central motor areas, reflecting increased motor preparation for temporally predictable targets. Second, on the neural network level, respiration-modulated beta-band activity within anterior cingulate cortex (ACC), supplementary motor area (SMA), and visual cortices (V1/2) predicted individual perceptual sensitivity. Third, this link between respiration, ACC beta oscillations, and behaviour was dependent on target predictability: When cueing information was available, respiration-beta coupling was more predictive of perceptual performance than when cues were absent. Beta-band activity within

the ACC as part of the medial prefrontal cortex and a well-established site of respiration-modulated oscillations[2,22] has frequently been related to top-down signalling of sensory predictions[49].

Hence, we overall found respiration-modulated suppression effects in alpha and beta frequency bands, both of which were tightly linked to behaviour: Alpha suppression over visual cortices was predictive of trial-by-trial stimulus detection (hit vs miss) whereas beta suppression within ACC, SMA, and visual cortex was related to perceptual threshold. In the context of predictive processing, ongoing beta-band activity has been proposed as a correlate of prior beliefs in motor control schemes[47] reflecting precision weighting of sensory prediction errors both prior to and following a movement[50].

Collectively, our findings of prestimulus visual alpha and central beta suppression during the inspiratory phase underscore the vital influence of the breathing rhythm on the neural processes which govern human perception. Together with the behavioural evidence outlined above, the tight link of these respiration-related excitability modulations to sensory sampling as well as their flexible adaptation to predictability changes strongly suggest that the respiratory rhythm is adaptively aligned with task dynamics in order to form an increasingly precise predictive model of upcoming sensory information and optimally prepare corresponding behavioural responses. To this end, sensory sampling is temporally coordinated with upstates in excitability and arousal dynamics to facilitate perception.

One key insight from the present work concerns the functional mechanism by which this facilitative coupling of brain and body is achieved on the neural network level. Based on a previous comprehensive mapping of RMBOs across the human brain[2], we conducted respiration phase-resolved analyses of directed functional connectivity in order to unravel breathing-related changes in network communication. Across experimental conditions, respiration phase modulated broadband signal flow within the RMBO network, particularly in fronto-parietal sites comprising insula, ACC/PCC, TPJ, and primary as well as supplementary motor areas. Anterior cingulate and insular cortex are at the core of the so-called interoceptive network, a set of cortical and subcortical nodes consistently reported to be responsible for interoceptive signalling. According to the embodied predictive interoception coding model proposed by Barrett and Simmons[51], ACC and INS jointly encode top-down interoceptive predictions and integrate bottom-up prediction errors. Fittingly, neural oscillations within the adjacent PCC have also been shown to be coupled to respiration[2,5]. While visceromotor cortices (including M1) have been proposed to drive active interoceptive perception in the brain[52], supplementary motor cortex (SMA) is of particular interest in the context of respiratory modulations: SMA is known to contain representations of respiratory muscles[53], which is why it has long been implicated in respiratory control. At the same time, it is intricately connected to both (pre-)frontal cortices (like ACC) and the anterior thalamus[54], other core nodes of interoceptive processing, and critical for relaying respiratory signals from the brainstem to the cortex. In our analyses, respiration-driven changes in functional connectivity between this set of nodes spanned a wide range of frequencies, indicating a global peripheral influence on both feedforward and feedback network processing. We observed stronger respiratory modulation between visual cortices and TPJ whenever participants were able to predict upcoming stimuli. Additionally, stronger alpha-band connectivity connectivity from the insular cortex to SMA was associated with higher perceptual sensitivity for these targets. Neural activity within TPJ is modulated by the respiratory rhythm[2,5] and the region has long been established as part of a ventral attentional control network[55]. Geng and Vossel[56] further proposed contextual updating as one main computational role of TPJ in cognitive processing, i.e. the updating of internal models about current context characteristics based on new sensory information. Our finding that prestimulus V1/2 → TPJ connectivity was indicative of perceptual sensitivity in predictable

contexts very likely reflects contextual updating: The internal model of the present probabilistic context is updated within TPJ from visual input allowing temporal and spatial prediction of an upcoming stimulus. Furthermore, during highly predictable trials, the insular cortex exhibited higher connectivity with a motor-control area (SMA). The insular cortex is arguably the most important structure in interoceptive predictive processing[7]. The insula receives a vast amount of interoceptive information from various visceral organ systems to keep track of the body's internal state (e.g., heart rate, respiration, visceral sensations[57]). Based on these multifaceted peripheral inputs, the insula plays a central role in interoceptive attention, i.e. the conscious perception and interpretation of these internal bodily signals. Critically, the insula not only receives bodily signals for the purpose of monitoring homoeostasis, but rather acts as an integrative hub, bridging internal physiological states with external environmental stimuli[58]. As such, the insula has been put to the forefront of predictive processing accounts of brain-body interactions. The Insula Hierarchical Modular Adaptive Interoception Control model (IMAC[59]) highlights the insula as a crucial cortical region for hierarchical predictive processing of internal bodily states, enabling the brain to anticipate, regulate, and adapt physiological responses to environmental and bodily demands. In our case, results suggest that the insular integrates external information about the current context (i.e., a predictable upcoming stimulus), integrates it with internal information (current respiratory state), and conveys this information to motor control for subsequent action. During the rather long prestimulus interval, respiration can then be voluntarily adjusted to receive the near-threshold stimulus during the inspiratory phase, thus maximising the likelihood of perceiving it thanks to the inspiration-associated upregulation of cortical excitability.

Overall, we observed that the respiratory rhythm dynamically modulated sensory sampling through changes in directed functional connectivity in a network governing interoceptive inference. This network appeared to be anatomically stable and centred on integrative hubs like TPJ and insula, yet with a certain specificity regarding partnering regions and their role in behaviour: Investigating respiration-related oscillatory modulations, we observed that baseline changes in beta power within SMA and V1/2 covaried with perceptual sensitivity (Fig. 3). Correspondingly, our respiration phase-resolved Granger causality analysis revealed SMA and V1/2 as the partner regions of insular and temporoparietal hubs which reflected alpha-band connectivity changes as a modulator of individual perceptual sensitivity (Fig. 4). In a final step, it is therefore highly instructive to consider low-frequency neural oscillations as the way of communication within that network: In frameworks of predictive processing, neural computations are hypothesised to reflect the process of continuously revising top-down predictions in response to bottom-up sensorimotor prediction errors. The interplay of feedforward and feedback computations gives rise to neural oscillations whose hierarchically organised frequencies encode neural communication. As outlined above, we observed connectivity changes between V1/2 → TPJ and perceptually relevant TPJ → INS as a function of target predictability across low frequencies, with strongest effects in the alpha band (around 8 Hz). We interpret these rhythms as top-down propagation of internal predictive models in theories of predictive processing[60]. Note that any directional connectivity analysis cannot strictly exclude influences like source leakage or infraslow modulation. Hence, even the compelling phase-related directionality changes presented here reflect statistical predictability rather than immediate anatomical or causal direction.

To summarise, in this study we have shown that respiration is a powerful and flexible modulator of human perception. Our results reveal a multi-layered mechanism underlying brain-body interactions during sensory sampling: Individuals voluntarily adapt their breathing patterns to task demands, a strategy that enhances perceptual sensitivity by aligning sensory sampling with phases of optimal cortical excitability and arousal. These states are reflected in the systematic modulation of alpha and beta oscillations, which are linked to visual processing and motor readiness, respectively. At the network level, we have identified that respiration shapes directed information flow within a key interoceptive system, with connectivity from the temporoparietal junction to the insula playing a crucial role in adapting to predictable sensory contexts. This work solidifies the role of respiration as a cornerstone of active sensing in humans and provides a critical foundation for future studies examining how this vital brain-body dialogue is implemented and how it might be leveraged to understand and potentially ameliorate clinical conditions, as we and others have previously suggested[37,38,61].

Since brain-body neuroscience research (and the associated field of interoception) continues to be of great interest for public and medial discourse, it is important to state limitations regarding the interpretability and 'everyday' applicability of our findings. In order to test our hypotheses, we used a great amount of experimental manipulation and very specific near-threshold stimuli. Overall, this task context constitutes a highly controlled regime which does not immediately translate to the experience of daily, natural sensing. We have recently argued the importance of fundamental research and mechanistic insight before applications or interventions can be derived from singular findings[61], and the very same cautionary note applies here as well. Further research is needed in order to unravel the complexities of the brain-body axis, particularly with regard to interactions across organ systems. Respiratory and cardiac dynamics, for example, are tightly coupled through respiratory heart rate variability[62]. However, this link by no means excludes the possibility of complementary pathways, particularly because recent findings point towards independent modulations of corticospinal excitability by respiratory and cardiac rhythms[63].

Methodologically, further variations of perceptual tasks could explore the present effects not only across sensory domains (e.g., in audition), but also with variable timings during the prestimulus fixation period for more precise readouts of individual respiratory alignment. Moreover, the benefit of catch trials (i.e., no stimuli presented) could be exploited to compute established signal detection theory measures like false alarm rate and decision criterion.

## Methods

### Participants

Thirty right-handed volunteers (15 female, age 25.9 ± 3.3 y [mean ± SD]) participated in the study. Self-reported sex or gender of participants did not play a role in the experimental design of the study. All participants reported having no respiratory or neurological disease and gave written informed consent prior to all experimental procedures. The study was approved by the local ethics committee of the University of Münster (approval ID 2018-068-f-S).

### Procedure

Participants were seated upright in a magnetically shielded room while we simultaneously recorded respiration and MEG data. MEG data were acquired using a 275-channel whole-head system (OMEGA 275, VSM Medtech Ltd., Vancouver, Canada) and continuously recorded at a sampling frequency of 600 Hz. To minimise head movement, the participant's head was stabilised with cotton pads inside the MEG helmet. Data were acquired across six runs with intermediate self-paced breaks. The length of each run was dependent on individual response speed (group-level mean ± SD: 452 ± 28 s). Participants were to breathe automatically through their nose while respiration was recorded as thoracic circumference by means of a respiration belt transducer (BIOPAC Systems, Inc., Goleta, United States) placed around their chest. Participants were continuously monitored via video to ensure they kept their mouths closed and were breathing through their noses.

## Task

Participants performed a spatial detection task (3-choice, single interval) in which they were to detect peripheral visual stimuli presented on a black computer screen. Specifically, each trial comprised a fixation period (1600 ms) during which participants were to fixate on the centre of the screen. After this fixation period, a small Gabor patch (0.3 degrees in diameter) was presented for 50 ms in one of two marked circular areas (3.5 degrees in diameter) located 10 degrees to the left or the right side of the fixation cross. Following a delay of 500 ms, a question mark in the centre of the screen prompted participants to give their response: Using a 3-button response box in their right hand, participants reported whether they saw a target on the left side (index finger), on the right side (middle finger), or no target at all (thumb). Once the report was registered, a new trial started (again with a fixation period). Critically, the predictability of target stimuli was manipulated along two dimensions (Fig. 1): During the fixation phase, participants were shown either a neutral fixation cross (uncued trials, C-) or an arrow head indicating the spatial location of the upcoming target (cued trials, C + ). In addition to this spatial cueing, trials could be timed (T + ) by a visual countdown surrounding the central fixation point which - in contrast to untimed (T-) trials - allowed participants to temporally predict the onset of the target stimulus. Independent manipulation of cueing and timing resulted in a full 2 × 2 design as shown in Fig. 1a. Each combination (C + T + , C + T-, C-T + , C-T-) was presented 30 times per experimental run ($n = 120$ trials per run), yielding 180 trials per condition over the entire experiment. Target locations (left, right) were balanced within each condition and experimental run. For each trial, target contrast was adapted by a QUEST staircase[64] aimed at individual overall hit rates of about 60 %. Prior to the first run, participants were instructed to keep their eyes on the centre cross at all times and encouraged to report left or right targets even when they were not entirely certain. Participants were told that, in addition to potential left and right targets, there were trials where no target was presented at all. In fact, no such catch trials occurred. Note that, while this was an important and deliberate decision in designing our task, it comes at the cost of certain SDT metrics (e.g., false alarm rate) not being able to be computed. Finally, participants were instructed that the spatial cue indicated the target location "with a certain probability" so that the target could still appear at the opposite location or be omitted altogether. Again, no such trials occurred and the spatial cues were in fact deterministic. Participants were debriefed at the end of the MEG session and reported not to have noticed the deterministic nature of the spatial cues.

## Resting state

For each participant, two sessions of resting state data were collected (six minutes each). In one session, participants breathed naturally through their noses while being video-monitored to confirm that their mouths remained closed. During the other session, participants were instructed to breathe through their mouth while a nose clip was applied to block nasal airflow. The order of nasal and oral breathing sessions was counterbalanced across participants.

## MRI acquisition and co-registration

For MEG source localisation, we obtained high-resolution structural magnetic resonance imaging (MRI) scans in a 3 T Magnetom Prisma scanner (Siemens, Erlangen, Germany). Anatomical images were acquired using a standard Siemens 3D T1-weighted whole-brain MPRAGE imaging sequence (1 × 1 ×1 mm voxel size, TR = 2,130 ms, TE = 3.51 ms, 256 × 256 mm field of view, 192 sagittal slices). MRI measurement was conducted in the supine position to reduce head movements, and gadolinium markers were placed at the nasion as well as left and right distal outer ear canal positions for landmark-based co-registration of MEG and MRI coordinate systems. Co-registration of structural T1 MRIs to the MEG coordinate system was done for each

participant by initial identification of three anatomical landmarks (nasion, left and right pre-auricular points) in their individual MRI. Using the implemented segmentation algorithms in Fieldtrip and SPM12, individual head models were constructed from anatomical MRIs. A solution of the forward model was computed using the realistically-shaped single-shell volume conductor model[65] with a 5 mm grid defined in the Human Connectome Project (HCP) template brain[31] after linear transformation to the individual MRI.

## Behavioural analyses

Behavioural data were preprocessed and analysed using Matlab (The Mathworks, Inc., Natick, United States). To account for the initiation of the QUEST procedure, the first 10 trials of each run were discarded. Trials were classified into hits (i.e., detected targets) and misses (undetected targets). Individual hit rates (HR) were computed as $n_{hits}/(n_{hits} + n_{misses})$. For respiration phase-locked behavioural analyses, we used the respiration phase vectors described above to obtain the respiration phase angle corresponding to fixation onset and target onset of each trial.

## Respiration phase-dependent fitting of the psychometric function

To quantify respiration phase-locked changes in perceptual accuracy, we employed a robust Bayesian inference analysis for the psychometric function. For each participant, we first fitted the psychometric function (with a cumulative Gaussian distribution as the sigmoid) to all target trials using a three-alternative forced choice model from the Psignifit toolbox[66] for Matlab. We then used the identified threshold and slope from the overall fit as priors for an iterative refitting of the psychometric function on subsets of trials obtained from a moving window across respiratory phase: Moving along the respiration cycle in increments of $\Delta\omega = \pi/30$, we fitted the psychometric function to the trials presented at a respiration angle of $\omega \pm \pi/10$. This way, we extracted respiration phase-dependent threshold estimates for each participant (Fig. 3e) which were subsequently z-scored. Finally, the normalised threshold variations were averaged across participants to obtain the grand average phase-dependent threshold modulation at the population level.

## MEG preprocessing and segmentation

All MEG, respiratory, and pupil data preprocessing was done in Fieldtrip[67] for Matlab. As a first step in preprocessing, we adjusted the synthetic gradiometer order to third order to improve MEG noise balancing (ft_denoise_synthetic). Power line artefacts were removed using a discrete Fourier transform (DFT) filter on the line frequency of 50 Hz plus harmonics (including spectrum interpolation; dftfilter argument in ft_preprocessing). Next, bad channels were identified semi-automatically by means of amplitude and kurtosis criteria and subsequently repaired with cubic spline interpolation. For kurtosis, we applied a general threshold of $\omega > 10$ and outliers in amplitude were judged relative to all other sensors by an experienced analyst (DSK). We then used independent component analysis (ICA) to capture eye blinks and cardiac artefacts (ft_componentanalysis with 20 extracted components). On average, artefacts were manually identified in $1.50 \pm 0.42$ components (M ± SD) per participant and removed from the data. Finally, continuous MEG data were segmented to a [−1.85 0.75] s interval around target onset and resampled to 300 Hz.

## Respiration preprocessing and phase extraction

Raw respiration time courses were resampled to 300 Hz and z-scored to yield normalised respiration traces for each participant and each experimental run. To account for occasional, unusually high-amplitude breaths (e.g., sighs), segments that exceeded a normalised amplitude of $z = \pm 2.5$ were linearly interpolated (i.e., clipped). This way, such artefacts did not bias the peak detection algorithm we

subsequently applied to identify time points of peak inspiration (peaks) and peak expiration (troughs; findpeaks function in Matlab with minimal peak prominence set to 1). In addition, individual respiratory time series were visually inspected for noticeable breath holds, but none were detected. Continuous respiratory phase angles were linearly interpolated from trough to peak (−π to 0) and peak to trough (0 to π) in order to yield respiration cycles centered around peak inspiration (i.e., phase 0). Individual breathing rates were extracted as the mean distance between inspiratory peaks as defined by the peak detection algorithm described above.

## Pupil preprocessing
Pupil data were preprocessed following procedures of previous work[18]. In short, area traces were converted to pupil diameter to linearize our measure of pupil size. Blinks were identified and visually validated by an automatic procedure (available from https://github.com/anne-urai/pupil_preprocessing_tutorial) and linearly interpolated. Blink-interpolated pupil time series were subjected to the procedure again, using a relaxed criterion of $z = 6$ SD to capture remaining artifacts. Next, canonical responses to blinks were estimated and removed from the pupil time series. To that end, pupil time series were band-pass filtered (pass band: 0.01–10 Hz, second-order Butterworth, forward-reverse two-pass). For subsequent analyses, pupil diameter time series were converted to z scores using a robust procedure (MATLAB function 'normalise' with options set to 'zscore' and 'robust').

## Extraction of MEG time series in source space
For source-space MEG analyses, we extracted neural time series from a total of $n = 230$ cortical parcels from the HCP atlas[31]. Source reconstruction was performed using the linearly constrained minimum variance beamformer approach (LCMV[68]), where the lambda regularisation parameter was set to 5%. This approach estimates a spatial filter for each location of the 5-mm grid along the direction yielding maximum power. A single LCMV beamformer[68] was used to estimate the parcel-level time series across all trials and conditions. Parcels of interest were selected from the original publication in which we had established the RMBO network[2] based on the same atlas.

## Head movement correction
In order to rule out head movement as a potential confound in our analyses, we used a GLM-based compensation procedure established by Stolk and colleagues[69]. This method uses the accurate online head movement tracking that is performed by our acquisition system during MEG recordings. This leads to six continuous signals (temporally aligned with the MEG signal) that represent the x, y, and z coordinates of the head centre ($H_x$, $H_y$, $H_z$) and the three rotation angles ($H_\psi$, $H_\theta$, $H_\varphi$) that together fully describe head movement. We constructed a regression model comprising these six 'raw' signals as well as their derivatives and, from these 12 signals, the first-, second-, and third-order non-linear regressors to compute a total of 36 head movement-related regression weights (using a third-order polynomial fit to remove slow drifts). This regression analysis was performed on single-parcel time courses after source reconstruction (see above), removing signal components attributable to head translation or rotation relative to the MEG sensors.

## Spectral decomposition and connectivity analysis
We used a non-parametric approach to estimate pairwise multivariate Granger causality between each pair of ROIs. The spectral representation of the source-localised time series was estimated using the fast Fourier transform along with multitapers (using 2-Hz smoothing) on the time domain data (separately for each condition) for the time interval of [−1600 0] ms relative to target onset. For each pair of ROIs we computed the cross-spectral density matrix and used it to compute multivariate Granger Causality (mGC) using a nonparametric spectral

matrix factorisation in a blockwise approach[70]. For this, we considered the first three principal components of each ROI's source-localised, trial-based time series. After obtaining the mGC spectra, we computed the directed asymmetry index (DAI[34]) as follows:

$$DAI = \frac{mGC(A \to B) - mGC(B \to A)}{mGC(A \to B) + mGC(B \to A)}$$

This procedure was repeated iteratively for a subset of trials corresponding to separate bins of respiration phase angle at the target onset (30 non-overlapping bins covering a full cycle of inspiration and expiration).

## Statistical analysis
Group-level significance was evaluated using cluster-based permutation tests implemented via Fieldtrip's ft_freqstatistics function. Initially, two-tailed t-tests were performed for each data point (representing power, mGC, DAI, or regression coefficient values) within every sensor or ROI; either comparing experimental conditions, or comparing an experimental condition against zero (this was applicable only for the DAI values). Resulting t-values were thresholded at $p = 0.05$. Spatially (for sensor space) and spectrally adjacent significant data points were grouped into clusters. We defined each cluster's summary statistic as the sum of absolute t-values and we randomly permuted condition labels 5000 times to generate a null distribution (note that for each permutation we recomputed initial t-values, redetected clusters, and recalculated cluster-level statistics). Then, original cluster statistics were compared against the permuted null distribution. Clusters exceeding the 97.5th percentile of this distribution ($p < 0.05$) were deemed statistically significant.

## Reporting summary
Further information on research design is available in the Nature Portfolio Reporting Summary linked to this article.

## Data availability
Raw data are protected by data privacy laws and can be made available by means of a formal data sharing agreement. The processed and anonymised data MEG, physiological, and behavioural data are publicly available at the Open Science Framework (OSF) via osf.io/qasvp. Source data are provided with this paper.

## Code availability
Software code is publicly available via osf.io/qasvp and included in the source data folders provided with this paper.

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

## Acknowledgements

NC is supported by the DFG (grant ID 491157081). DSK is supported by the European Union (ERC StG No. 101162169), the Innovative Medical Research (IMF) programme of the University of Münster (grant ID KL 1 2 22 01), and the IZKF programme of the University of Münster (grant ID Klu3/002/26).

## Author contributions

Conceptualization, D.S.K.; Methodology, N.C., D.S.K.; Investigation, D.S.K., M.S., T.B.; Writing – Original Draft, N.C., D.S.K.; Writing – Review & Editing, all authors; Visualisation, N.C., D.S.K.; Funding Acquisition, D.S.K.

## Funding

## Competing interests

The authors declare no competing interests.
