## [Transparent Peer Review file · Nature Communications]

Respiration as a dynamic modulator of sensory sampling

Corresponding Author: Dr Daniel Kluger

Version 0:

Reviewer comments:

Reviewer #1

(Remarks to the Author)

This manuscript presents work in which the link was tested between respiration, arousal, neural oscillations, connectivity and how together they impact behaviour as measured in a perceptual sensitivity task. The experiment is carefully set up to answer the research questions and all analyses are solid and supported by the underlying hypotheses. The work demonstrates on the behavioural level how respiratory phase and alignment of breathing can change detection thresholds and how these effects are directly linked to changes in pupil size. Furthermore, these results are then extended to show that in both the visual and motor domain, respiration couples to alpha and beta oscillations in a behaviorally meaningful way. Lastly, using Granger causality analyses, the authors show how respiration influences directed information flow within a set of ROIs of the RMBO. The work is well done and the manuscript clearly written, so I have few comments to mostly improve the accessibility of the paper.

Overall, the results section could benefit from explaining the implemented analyses in a way that is more accessible to a broad readership, given that there is no dedicated analysis section in the manuscript. I give some examples below on the first part concerning the behavioural results, but the logic can be extended to the other sections as well:

- Page 4: please clarify “.. we also included the sine and cosine of the prestimulus respiratory phase”. Respiratory phase at which moment in the task exactly? Why were sin and cos taken used?
- The description of refitting the psychometric curve of the behavioural data using resp. phase bins (i.e. moving window approach) could do with some rewriting, to make the logic/implementation of the analysis overall more clear.

In the methods section:

- There is no description of the acquisition of resting state data, however RR rates were compared to rest in the analyses. It would be good to describe the acquisition of the resting state data in terms of duration, general instructions and breathing mode.

- For the MEG preprocessing: were components identified manually or automatically?

- Was there a specific criterium for bad channel identification? The current description of using amplitude and kurtosis is difficult to replicate based on the text alone.

Typos:

- Page 4: 'peristimulus'

Reviewer #2

(Remarks to the Author)

Chalas and colleagues provide valuable new insights into the interplay between brain dynamics, perception, and respiration. Building on their 2021 study, the authors use a near-threshold visual detection task to probe perceptual sensitivity across the respiratory cycle, while recording arousal via pupillometry and brain activity with MEG. The task manipulates stimulus predictability (very nice) with spatial and temporal cues.

Methods are sound and reproducible; the use of robust statistics (e.g., cluster-based permutation tests) is appropriate. Results are reported carefully and target mechanisms rather than correlations. Beyond confirming higher sensitivity during mid- to late inspiration, a clear novelty is shown: directional connectivity indicates increased low-frequency influence from temporoparietal junction (TPJ) to insula prior to target onset is associated with higher perceptual sensitivity. The Discussion situates these findings within an interoceptive predictive-processing account in which greater predictability enables tighter coupling across systems (respiration, beta suppression, pupil dilation). Within this view, TPJ-to-insula influence is

interpreted as contextual updating that informs interoceptive control to optimise perceptual sensitivity. Overall, this is a careful and innovative manuscript with a substantive contribution.

I recommend revisions aimed at clarifying scope, improving mechanistic transparency and the framing.

General framing and theory:

Situate respiration within active sensing taxonomies. Add two sentences in the Introduction to acknowledge prior distinctions between alloactive sensing (sensor reconfiguration; e.g., eye movements, posture, respiration pacing) and homeoactive sensing (energy injection; e.g., palpation, echolocation). Note that respiration is a partly volitional, predominantly alloactive control that can reshape the internal milieu supporting intake. Provide at least one representative citation to this literature and clarify how your task engages alloactive control. See Zweifel, N. O., & Hartmann, M. J. (2020)

Temper generalisation to everyday sensing:

Add a brief statement that near-threshold, brief, externally timed Gabor stimuli capture a specific, informative regime, but are not representative of naturalistic, sustained, symbolic, or self-paced sensing. You already emphasise predictability manipulations; explicitly note that effects may be smaller or policy-expressed rather than sensitivity-expressed in ecological tasks. Place this either in Discussion or a short "Limitations and scope" note. In short, it would be nice to directly state that the current experiments represent one portion of natural sensing possibilities in daily experiences; so further researchers/media don't oversell the mechanism.

Acknowledge cardio-respiratory coupling as a complementary pathway:

Add some sentences noting that respiration can modulate cardiac timing via respiratory sinus arrhythmia, thereby co-shaping permissive windows for perception, and that concurrent ECG recording would allow testing whether part of the observed effects are mediated by changes in cardiac phase or heart-breath coupling. At present ECG does not appear to be recorded; stating this as a limitation clarifies mechanisms. See Menuet et al., 2025

Motor-related confounds for beta modulation:

Because responses are manual each trial and beta power indexes motor readiness, please clarify how you separated sensory-related beta effects from motor preparation.

Criterion vs sensitivity under a 3-choice task without catch trials:

You state no catch trials occurred although participants had a "no-target" button. This design prevents standard SDT estimation of false alarms and decision criterion. Please justify this choice, clarify how misses and "no-target" presses were treated, and consider reporting complementary bias-robust metrics. A brief note in Methods and maybe a sentence in Limitations would suffice.

Consistency with mixed respiratory-phase findings:

You report inspiratory facilitation. Some studies report exhalation-related benefits depending on modality, task, and sampling constraints. Add two cautious sentences in the Discussion outlining boundary conditions under which inspiration versus exhalation may be optimal, and relate this to predictability and the ability to prolong sampling. This will preempt over-generalisation.

Presentation:

Given the many pairwise relations reported (pupil, phase, threshold, alpha, beta), a compact diagram indicating putative facilitatory or inhibitory links and approximate magnitudes would help readers follow the mechanistic story. This is a suggestion, not a requirement.

Breath holds and respiratory phase mapping in circular statistics:

Please clarify how respiratory phase was derived and mapped. In many pipelines inhalation and exhalation are each assigned 180°, yet inspiratory and expiratory durations typically differ within and between participants, and brief breath holds are common. This asymmetry can bias phase-resolved effects if time spent in each phase is not represented accurately. Plus, State how holds and pauses were detected and handled (e.g., excluded, merged with exhalation, or treated as a distinct category). Were these even present?

Overall, We offer a favourable evaluation. The study is careful, the methodology is innovative, and the TPJ-to-insula connectivity result adds a novel piece to the respiration-brain-perception puzzle. With the minor clarifications and additions above, the manuscript will provide a stronger bridge between near-threshold laboratory paradigms and broader active-sensing accounts of respiration, brain dynamics and vision.

Reviewer #3

(Remarks to the Author)

1. What are the noteworthy results?

This study provides compelling evidence that respiration actively shapes human perception rather than serving as a passive, automatic background rhythm. Chalas et al. show that individuals fine-tune their breathing patterns to optimize sensory performance, aligning inhalation phases with moments of heightened cortical excitability and arousal preceding predictable stimulus onset. Corresponding shifts in alpha- and beta-band activity, measured with MEG, indicate that breathing modulates both visual processing and motor readiness to near-threshold visual stimuli. At the network level, directed connectivity from the temporoparietal junction to the insula reveals a key interoceptive pathway through which

respiration influences perceptual context. Overall, the work positions breathing as a fundamental element of active sensing in humans and provides a framework for understanding - and ultimately treating - disorders of perception, attention, and interoception.

2. Will the work be of significance to the field and related fields? How does it compare to the established literature? If the work is not original, please provide relevant references.

It adds on the existent knowledge about respiratory-related perceptual modulation by more precisely specifying how respiration modulates neural excitability and arousal to aid performance on a perceptual sensory task. Previous papers (e.g., Perl et al 2019) have shown similar effects but this work takes it a step further.

3. Does the work support the conclusions and claims, or is additional evidence needed?

It supports the conclusions and claims but see below.

Answers to the questions below is required.

4. Are there any flaws in the data analysis, interpretation and conclusions?

See below

Do these prohibit publication or require revision?

They do not prohibit publication but consideration. See below

5. Is the methodology sound? Does the work meet the expected standards in your field?

Yes

6. Is there enough detail provided in the methods for the work to be reproduced?

Yes

Remarks to the authors related to points 3 and 4 above.

This study employs a variation of the classic Posner task, a well-established paradigm for probing visuospatial attention. This is a strong choice to demonstrate the role of respiration in perceptual processing, especially through the use of both spatial and temporal cues and well-controlled stimuli (Gabor patches). The fixed 1.6-s interval before target onset was intentionally designed to allow participants to align their respiratory phase with the upcoming stimulus. However, this design also introduces potential limitations.

If respiratory modulation were such a strong enhancer of perceptual sensitivity, one might expect participants—particularly those with better temporal alignment—to start adjusting their breathing even before cue presentation. Because the 1.6 s interval was constant, predictable, and relatively short in respiratory terms, an optimal strategy could have been to reset breathing as soon as the fixation cross appeared, ensuring that a favorable phase angle recurred 1.6 s later regardless of cue appearance. In this scenario, cues could serve to fine-tune performance rather than drive large respiratory phase shifts, which occur on a slower timescale.

Although many conclusions rely on within-subject analyses (e.g., trials with stronger phase alignment showing lower detection thresholds), this fixed-interval predictability remains a concern. A less predictable design—or inclusion of a control condition with jittered pre-target intervals (e.g., 1–6 s)—would have provided a stronger test of the hypothesis. Alternatively, an entrainment paradigm presenting streams of visual stimuli with variable inter-stimulus intervals and target probabilities (e.g., Lakatos et al., 2008) would be well suited to examining respiratory entrainment in an inherently rhythmic active-sensing context. While the use of temporal cues partly addresses this issue, the effect was modest ($t = 2.0$, $p = 0.047$) compared with the spatial cue benefit ($t = 4.28$, $p < 0.001$), suggesting that participants derived limited advantage from temporal information under the fixed 1.6 s schedule. The result on pre-target pupil modulation (fig 1h) also supports the claim that respiration adjusts in the absence of cues (larger pupil changes in C-T- than C+T-). In addition, the lack of condition X respiration phase interaction on directed RMBO network connectivity (fig 4g) also supports this claim.

Regarding hit rates, figure 1b shows that C+T+ is higher than C-T- and that C-T+ is higher than C-T-. However, C+T- looks higher than C-T- and maybe even higher than C-T+ also. Are these effects significant?

Fig 1g:

What are the units in the y-axis (pupil diameter)? Why they start between .2 and .3? If deflections downwards mean pupil constriction, then temporal cues (T+) constrict the pupil during the pre-target period up to target onset. This effect is even larger when temporal cues are presented alone (larger constriction in C-T+ compared to C+T+). Not sure this is clearly explained. Lighting was controlled across conditions discarding luminance changes. Were there fixation differences across conditions (e.g., more saccades to temporal cues in C-T+ than C+T+) or is the effect explained by arousal differences? Why C-T- shows more pre-target pupil constriction than C+T-?

Previous studies have shown that during spontaneous ventilation (no stimuli) pupil size expands during inhalation and contracts during exhalation. Namely it starts expanding 1-1.5s after inhalation onset. Are the present arousal-related pupil changes correlated to the subsequent inhalation-related pupil changes after target onset? Given that in this study, most subjects align their mid/late inhalations to target onset to improve detection, were there larger inhalation-related pupil changes in resp aligned trials?

“Controlling for lighting changes on the screen, spatial cues significantly modulated arousal during prestimulus fixation leading up to target presentation, irrespective of temporal cues (FDR-corrected $p < .05$; Fig. 1g).” Does that mean that spatial cues prevent pupil constriction from happening during the pre-target period?

“Pupil diameter at target onset was systematically related to respiration phase at target onset (Rayleigh test of individual circular means: $z(29) = 8.51$, $p < .001$), with greater diameter indicating higher arousal during inspiration”. Is the greater pupil constriction in C-T+ compared to C+T+ at pre-target and target onset related to higher arousal due to higher uncertainty about the target?

Using a mutual information model, the authors showed that near-threshold visual detection can be predicted not only by spatial-temporal cues, but also by the respiratory phase at stimulus onset. Incorporating the sine and cosine of the peristimulus breathing cycle significantly improved single-trial prediction accuracy, demonstrating that both stimulus predictability and ongoing respiratory state jointly shape perceptual outcomes. This analysis robustly replicates and extends prior reports of respiration-linked perceptual modulation.

It is hard to grasp some of the aspects in Fig 1f. Black dots seem quite widely distributed in the y-axis. Does this mean that preferred resp phase is different in each subject? This interpretation contrast with fig 1c – in which mid inhalation is the ideal timing across all subjects (e.g., lower contrast gabor patches are perceived if presented during mid inhalation) and fig 1e. Maybe showing the raw or averaged respiration waveforms overlaid would be helpful.

The result that respiratory rate during task is higher than rest could be linked to the fact that it is easier to make phase adjustments in short resp cycles compared to longer ones.

Regarding the analyses of the neural data, I believe the authors used state-of-the-art analyses and the results are strong with impressive correlations between neural activity in the RMBO network, performance and respiratory phase.

Given the activation of temporal site (TPJ)– did the participants heard their breathing sounds modulating auditory cortex activity? did they wear headphones?

It is known that hyperventilation can modulate neural excitability. Given that participants breathed faster during the task compared to rest, could this have influence some of their results? Was tidal volume measured or controlled in anyway?

Reviewer #4

(Remarks to the Author)

Version 1:

Reviewer comments:

Reviewer #2

(Remarks to the Author)

I would first like to thank the authors for their careful and thoughtful responses to the previous round of comments. The revised manuscript is substantially improved. The following points are offered as minor comments aimed at further strengthening clarity, scope, and interpretative precision. None of them detract from the overall quality or validity of the work.

Framing of active sensing and voluntary respiratory control: While the evidence for task-dependent respiratory adjustment and respiration-locked behavioural facilitation is compelling, some passages still read as slightly stronger than strictly warranted in terms of voluntary control. The data clearly support adaptive coupling between respiration, neural dynamics, and perceptual performance, but direct causal evidence for strategic respiratory control remains inferential.

It may therefore be helpful to slightly temper the language in places, e.g., by emphasizing adaptive alignment or task-dependent adjustment rather than fully intentional control.

Behavioural interpretation and perceptual sensitivity: Given the absence of catch trials, established signal detection theory measures such as false alarm rate and decision criterion cannot be computed. While this limitation is acknowledged later in the manuscript, it may be useful to briefly remind the reader earlier that respiration-related threshold effects could, in principle, reflect a combination of sensory sensitivity and response criterion shifts.

Related literature and completeness: For the sake of completeness, the authors may wish to consider citing a recent preprint that is closely related in scope and conceptual framing: https://osf.io/preprints/psyarxiv/zj23s_v1

Interpretation of directed connectivity analyses: The multivariate Granger causality analyses are carefully implemented and constitute a clear strength of the manuscript. That said, as with any GC-based approach, directed connectivity estimates may still be influenced by source leakage, common drivers, and slow rhythmic modulation.

A brief, explicit note acknowledging these general limitations, and clarifying that the reported directionality reflects statistical

predictability rather than anatomical or causal direction, would further strengthen interpretative caution and transparency. This would be particularly useful in sections linking specific INS–SMA or TPJ–V1 interactions to functional roles.

Density and presentation: The manuscript is necessarily rich, but some sections, particularly in the Results and Discussion, remain quite dense and occasionally reiterate similar points across behavioural, oscillatory, and connectivity levels. A modest reduction of repetition or slightly clearer "marking" of which analyses are confirmatory versus novel could improve readability without loss of content.

In summary, this is a strong and carefully executed study that makes a meaningful contribution to our understanding of respiration as a modulator of perception and neural dynamics. The points above are intended solely as minor refinements. I commend the authors for the quality of the work and for the clear improvements introduced in the revised version.

Reviewer #3

(Remarks to the Author)

The authors have answered my questions and now provide an improved manuscript. I support its publication.

Reviewer #4

(Remarks to the Author)

Response to Reviewers

We would like to thank all four reviewers for their insightful comments on our manuscript. We firmly believe to have included all their feedback in a revised version which, in doing so, has undoubtedly gained transparency and accessibility.

Below, we provide a point-by-point response to all reviewers with reviewer comments in **grey** and our responses in **blue**. Quoted changes to the manuscript are shown in **red**.

For the revised version of the manuscript we included two dedicated control analyses: one regarding the saccadic behavior prior to target onset and another regarding respiration modulation on pupil dilation. Both analyses strengthen our initial observation. Additionally, while revising our manuscript in accordance with the reviewers' comments, we found a bug in our code, related to undocumented functionalities in Matlab, which caused the Granger causality analysis to inadvertently account for one SVD component (instead of three) for half of the nodes (use of curly instead of regular brackets in one line). We recomputed all analyses reported in Fig. 4 and have revised the figure as well as the main text accordingly (marked in red). As a result of the new computations, we increased statistical power and now report a previously non-significant predictability condition effect between C+T+ and C-T- trials (see new Fig. 4h below). This new finding allows us to characterize consistent patterns of respiration phase-locked information flow within the RMBO network as a function of current statistical task context. Thanks to the increased sensitivity of the corrected analysis, this pattern unfolds much more clearly than before and reveals distinct sensory and motor-related subnetworks related to target predictability and perceptual accuracy, respectively.

Reviewer #1 (Remarks to the Author):

This manuscript presents work in which the link was tested between respiration, arousal, neural oscillations, connectivity and how together they impact behaviour as measured in a perceptual sensitivity task. The experiment is carefully set up to answer the research questions and all analyses are solid and supported by the underlying hypotheses. The work demonstrates on the behavioural level how respiratory phase and alignment of breathing can change detection thresholds and how these effects are directly linked to changes in pupil size. Furthermore, these results are then extended to show that in both the visual and motor domain, respiration couples to alpha and beta oscillations in a behaviorally meaningful way. Lastly, using Granger causality analyses, the authors show how respiration influences directed information flow within a set of ROIs of the RMBO. The work is well done and the manuscript clearly written, so I have few comments to mostly improve the accessibility of the paper.

We thank the reviewer for their positive evaluation of our manuscript; point-by-point responses to the reviewer's remarks are listed below.

Overall, the results section could benefit from explaining the implemented analyses in a way that is more accessible to a broad readership, given that there is no dedicated analysis section in the manuscript. I give some examples below on the first part concerning the behavioural results, but the logic can be extended to the other sections as well:

- Page 4: please clarify “.. we also included the sine and cosine of the prestimulus respiratory phase”. Respiratory phase at which moment in the task exactly? Why were sin and cos taken used?

We appreciate the reviewer's suggestion to provide more accessible explanations of our analysis pipeline and have rephrased the manuscript in several instances to include the proposed additions. While changes in the revised manuscript are listed below, we want to briefly elaborate on the inclusion on sine and cosine of respiration phase in our analysis: This approach is referred to as a circular GLM or harmonic regression and is a commonly used tool for quantifying the amount of variance (in a GLM) or shared information (in a mutual information analysis) explained by circular signals. In short, respiratory phase is made up of a sine and a cosine component, both of which are included separately as data vectors in the analysis (see Cremers and Klugkist et al., 2018). In a regression analysis, this yields beta weights for sine and cosine components of respiratory phase. Since the interpretation of sine or cosine alone is not meaningful, the vector norm of both beta weights is computed and tested against a surrogate distribution. Changes made to the manuscript are as follows:

In a second step, we also included sine and cosine of the respiratory phase at stimulus onset, which once again significantly improved the prediction of individual performance ($t(29) = 7.94, p < .001$). In sum, these results demonstrate that both stimulus predictability (spatial, temporal) and respiration phase - fully described by its sine and cosine components - had a significant effect on behavioural performance. (p. 4)

- The description of refitting the psychometric curve of the behavioural data using resp. phase bins (i.e. moving window approach) could do with some rewriting, to make the logic/implementation of the analysis overall more clear.

Thank you, the corresponding paragraph has been rephrased as follows:

We exploited single-trial information regarding stimulus contrast and detection performance to iteratively refit the psychometric function. **In short, the psychometric function expresses the sigmoidal relationship between the intensity of a stimulus and the probability of a particular response. In our analysis, we first fitted the overall psychometric function to all target trials (irrespective of condition), quantifying detection probability as a function of stimulus contrast.** The resulting parameters were used as priors for a moving window approach in which we iteratively refitted the psychometric function to a subset of trials presented at a certain range of respiration phase angles. **To this end, respiratory phase at the onset of each target stimulus was extracted and assigned to one of $k = 60$ phase bins with all other trials presented at a similar respiratory phase (see ⁴ and Methods for details). Refitting the psychometric function for the subsets of trials assigned to each of these overlapping phase angle bins, we thus obtained a threshold estimate characterizing perceptual performance at that respiration phase. (p. 4)**

In addition, a dedicated paragraph has been added to the Methods section to describe the refitting approach in greater detail (pp. 20-21):

Respiration phase-dependent fitting of the psychometric function. To quantify respiration phase-locked changes in perceptual accuracy, we employed a robust Bayesian inference analysis for the psychometric function. For each participant, we first fitted the psychometric function (with a cumulative Gaussian distribution as the sigmoid) to all target trials using a three-alternative forced choice model from the Psignifit toolbox ⁶⁴ for Matlab. We then used the identified threshold and slope from the overall fit as priors for an iterative refitting of the psychometric function on subsets of trials obtained from a moving window across respiratory phase: Moving along the respiration cycle in increments of $\Delta\omega = \pi/30$, we fitted the psychometric function to the trials presented at a respiration angle of $\omega \pm \pi/10$. This way, we extracted respiration phase-dependent threshold estimates for each participant (see Fig. 3e) which were subsequently z-scored. Finally, the normalized threshold variations were averaged across participants to obtain the grand average phase-dependent threshold modulation at the population level.

In the methods section:

- There is no description of the acquisition of resting state data, however RR rates were compared to rest in the analyses. It would be good to describe the acquisition of the resting state data in terms of duration, general instructions and breathing mode.

We agree with the reviewer. We added a 'resting state' paragraph in the Methods of the revised version of the manuscript. The paragraph now reads as follows (p. 20):

Resting state. For each participant, two sessions of resting state data were collected (six minutes each). In one session, participants breathed naturally through their nose while being video-monitored to confirm that their mouths remained closed. During the other session, participants were instructed to breathe through their mouth while a nose clip was applied to block nasal airflow. The order of nasal and oral breathing sessions was counterbalanced across participants.

- For the MEG preprocessing: were components identified manually or automatically?

The components were identified manually. This is now stated in the Methods section of the revised manuscript. The paragraph now reads as follows (p. 21):

On average, artefacts were manually identified in 1.50 ± 0.42 components ($M \pm SD$) per participant and removed from the data.

- Was there a specific criterium for bad channel identification? The current description of using amplitude and kurtosis is difficult to replicate based on the text alone.

Bad channels were identified through visual inspection by an expert analyst (DSK). Regarding amplitude, sensors were only interpolated when they were clear outliers relative to all other MEG sensors. For kurtosis, a general criterion of $kurtosis > 10$ was applied. The paragraph has been rephrased and now reads as follows (p. 21):

Next, bad channels were identified semi-automatically by means of amplitude and kurtosis criteria and subsequently repaired with cubic spline interpolation. For kurtosis, we applied a general threshold of $\omega > 10$ and outliers in amplitude were judged relative to all other sensors by an experienced analyst (DSK). We then used independent component analysis (ICA) to capture eye blinks and cardiac artefacts (*ft_componentanalysis* with 20 extracted components). On average, artefacts were manually identified in 1.50 ± 0.42 components ($M \pm SD$) per participant and removed from the data.

Typos:

- Page 4: 'peristimulus'

Thank you for reading our manuscript with great care - in this instance, 'peristimulus' in fact referred to the period *during* stimulus presentation, but has now been rephrased (see above).

Reviewer #2 (Remarks to the Author):

Chalas and colleagues provide valuable new insights into the interplay between brain dynamics, perception, and respiration. Building on their 2021 study, the authors use a near-threshold visual detection task to probe perceptual sensitivity across the respiratory cycle, while recording arousal via pupillometry and brain activity with MEG. The task manipulates stimulus predictability (very nice) with spatial and temporal cues.

Methods are sound and reproducible; the use of robust statistics (e.g., cluster-based permutation tests) is appropriate. Results are reported carefully and target mechanisms rather than correlations. Beyond confirming higher sensitivity during mid- to late inspiration, a clear novelty is shown: directional connectivity indicates increased low-frequency influence from temporoparietal junction (TPJ) to insula prior to target onset is associated with higher perceptual sensitivity. The Discussion situates these findings within an interoceptive predictive-processing account in which greater predictability enables tighter coupling across systems (respiration, beta suppression, pupil dilation). Within this view, TPJ-to-insula influence is interpreted as contextual updating that informs interoceptive control to optimise perceptual sensitivity. Overall, this is a careful and innovative manuscript with a substantive contribution.

Thank you, we appreciate the reviewer's favourable evaluation of our manuscript.

I recommend revisions aimed at clarifying scope, improving mechanistic transparency and the framing.

General framing and theory:

Situate respiration within active sensing taxonomies. Add two sentences in the Introduction to acknowledge prior distinctions between alloactive sensing (sensor reconfiguration; e.g., eye movements, posture, respiration pacing) and homeoactive sensing (energy injection; e.g., palpation, echolocation). Note that respiration is a partly volitional, predominantly alloactive control that can reshape the internal milieu supporting intake. Provide at least one representative citation to this literature and clarify how your task engages alloactive control. See Zweifel, N. O., & Hartmann, M. J. (2020)

We appreciate the reviewer's remarks regarding alloactive vs homeoactive sensing. As suggested, we have rephrased the Introduction which now reads as follows (p. 3):

At present, converging evidence from different lines of research strongly suggests respiratory involvement in active sensing, including adaptation of respiratory behaviour to task timing²⁸, modulation of excitability and arousal states⁴, and behavioural facilitation²⁹. In the animal literature, active sensing has been suggested to comprise both a *homeoactive* and an *alloactive* component³⁰. In this framework, alloactive sensing postulates the use of mechanical energy to change parameters of the sensory apparatus, which includes voluntary changes of the respiratory rhythm based on internal forward models of sensory predictions. What has so far been critically missing, particularly in humans, is a direct test of previous correlational findings in a dedicated predictive processing paradigm in order to answer fundamental open research questions: To what extent does respiration orchestrate

excitability and arousal states across different contexts of stimulus predictability? What are the behavioural benefits of this respiration-brain coupling? And finally, how is the facilitative coupling of brain and body mechanistically achieved on the neural network level?

An additional reference has been added to the Discussion section (p. 15):

Lending from *active sensing* accounts proposed in the animal literature ⁸, these accounts posit that multiple streams of sensory information - e.g., internal and external signals - are coordinated in a way that optimizes their integration and propagation ³⁵. **Since respiration is under voluntary control, it is a prime example of an internal signal by which the system can exert alloactive control over sensory inputs, e.g. by temporal alignment through dynamic changes in the pace of the respiratory rhythm we observed in the present study.** This *predictive timing* has been shown to maximise the signal-to-noise balance between interoceptive and exteroceptive processing ³⁶ and synchronise sensory sampling with favourable brain states of excitability ^{4,22} and arousal ¹⁸.

Temper generalisation to everyday sensing:

Add a brief statement that near-threshold, brief, externally timed Gabor stimuli capture a specific, informative regime, but are not representative of naturalistic, sustained, symbolic, or self-paced sensing. You already emphasise predictability manipulations; explicitly note that effects may be smaller or policy-expressed rather than sensitivity-expressed in ecological tasks. Place this either in Discussion or a short "Limitations and scope" note. In short, it would be nice to directly state that the current experiments represent one portion of natural sensing possibilities in daily experiences; so further researchers/media don't oversell the mechanism.

This comment is particularly appreciated, as brain-body research continues to be overinterpreted in public discussions and 'pop science' or medial contexts. As suggested by the reviewer, and to prevent overselling of our mechanisms, we have added a brief 'Limitations and scope' section at the end of the Discussion (pp. 18-19):

Since brain-body neuroscience research (and the associated field of interoception) continues to be of great interest for public and medial discourse, it is important to state limitations regarding the interpretability and 'everyday' applicability of our findings. In order to test our hypotheses, we used a great amount of experimental manipulation and very specific near-threshold stimuli. Overall, this task context constitutes a highly controlled regime which does not immediately translate to the experience of daily, natural sensing. We have recently argued the importance of fundamental research and mechanistic insight before applications or interventions can be derived from singular findings ⁶⁰, and the very same cautionary note applies here as well.

Acknowledge cardio-respiratory coupling as a complementary pathway:

Add some sentences noting that respiration can modulate cardiac timing via respiratory sinus arrhythmia, thereby co-shaping permissive windows for perception, and that concurrent ECG recording would allow testing whether part of the observed effects are mediated by changes in

cardiac phase or heart–breath coupling. At present ECG does not appear to be recorded; stating this as a limitation clarifies mechanisms. See Menuet et al., 2025

Thank you for raising the point of complementary or cross-modal brain-body axes, which is the main focus of an ongoing ERC project in our lab. Respiratory heart-rate variability (to comply with the terminology of Menuet and colleagues) is primarily driven by rhythmic changes in parasympathetic activity which inhibits cardiac signals and is strongest during expiration, resulting in a decrease in heart rate. Consequently, absolute heart rate changes between inspiration and expiration can again be voluntarily controlled through changes in the frequency (and, relatedly, depth) of breathing (Sroufe, 1971). The present key findings of respiratory adaptation facilitating perception strongly suggest that HRV- or cardiac phase-related modulations reported by colleagues in the field are at least partially explained by a respiratory influence. It is entirely unclear to what extent interactions between cross-frequency rhythmicities from different organ systems modulate neural activity and whether the single-modality contributions are additive or more complex in nature.

We acknowledge the reviewer's suggestion and have added cardiorespiratory interactions to the newly introduced Limitations section, which now reads as follows (pp. 18):

Further research is needed in order to unravel the complexities of the brain-body axis, particularly with regard to interactions across organ systems. Respiratory and cardiac dynamics, for example, are tightly coupled through respiratory heart rate variability ⁶¹. However, this link by no means excludes the possibility of complementary pathways, particularly because recent findings point towards independent modulations of corticospinal excitability by respiratory and cardiac rhythms ⁶².

Motor-related confounds for beta modulation:

Because responses are manual each trial and beta power indexes motor readiness, please clarify how you separated sensory-related beta effects from motor preparation.

Within a predictive processing framework, there is no need for a strict separation of sensory and motor-related components of respiration-related beta activity. Previous work from our lab and others has demonstrated respiration phase-locked modulation of beta activity in motor cortices (Kluger et al., 2021; Kluger et al., 2023) and of motor function (Kluger & Gross, 2021; Park et al., 2020). Interestingly, this link is also present in speech perception (Chalas et al., 2022) and in motor imagery, i.e. contexts in which no action has to be prepared because it is never executed (Park et al., 2022). Taken together, these findings make a strong argument for a more general role of beta power in the motor system. As for a potential confound, even if one would clearly distinguish motor readiness and sensory processing, the motor preparation component would be cancelled out in our statistical contrasts precisely because it is common across conditions.

Criterion vs sensitivity under a 3-choice task without catch trials:

You state no catch trials occurred although participants had a “no-target” button. This design prevents standard SDT estimation of false alarms and decision criterion. Please justify this choice, clarify how misses and “no-target” presses were treated, and consider reporting complementary bias-robust metrics. A brief note in Methods and maybe a sentence in Limitations would suffice.

The reviewer is correct in stating that false alarms and criterion cannot be computed in the absence of catch trials, which is why we refrain from doing so in the present manuscript. As a development from our initial eLife paper on a similar task (without the predictability manipulation), catch trials were removed in order to increase statistical power through a 50% higher trial count (720 trials instead of 480). Precisely because there were no non-target trials, the ‘no target’ response was counted as a miss. The Methods section as well as the newly added Limitations section have been amended as follows:

Participants were told that, in addition to potential left and right targets, there were trials where no target was presented at all. In fact, no such catch trials occurred. **Note that, while this was an important and deliberate decision in designing our task, it comes at the cost of certain SDT metrics (e.g., false alarm rate) not being able to be computed.** (p. 20)

Methodologically, further variations of perceptual tasks could explore the present effects not only across sensory domains (e.g. in audition), but also with variable timings during the prestimulus fixation period for more precise readouts of individual respiratory alignment. Moreover, the benefit of catch trials (i.e. no stimuli presented) could be exploited to compute established signal detection theory measures like false alarm rate and decision criterion. (p. 19)

Consistency with mixed respiratory-phase findings:

You report inspiratory facilitation. Some studies report exhalation-related benefits depending on modality, task, and sampling constraints. Add two cautious sentences in the Discussion outlining boundary conditions under which inspiration versus exhalation may be optimal, and relate this to predictability and the ability to prolong sampling. This will preempt over-generalisation.

This is closely related to the above comment regarding overgeneralisation (and equally appreciated); the Discussion has been amended and now reads as follows (p. 15):

As for brain-body involvement, the respiratory cycle has been shown to modulate perceptual decision thresholds, particularly during inhalation ^{2,9}. **Note, however, that some studies report expiratory facilitation of behaviour ^{6,40}, suggesting that the distinction between inspiration- and expiration-locked modulation may depend on whether the events of interest primarily concern sensory perception or an immediate behavioural response.** In the present study, respiration-modulated changes in alpha activity temporally preceded shifts in pupil dilation, indicating that arousal neuromodulation is primed in neural circuits and, as a result, affects perceptual processing ².

Presentation:

Given the many pairwise relations reported (pupil, phase, threshold, alpha, beta), a compact diagram indicating putative facilitatory or inhibitory links and approximate magnitudes would help readers follow the mechanistic story. This is a suggestion, not a requirement.

We thank the reviewer for proposing a way to help make the admittedly rich pattern of findings more accessible for readers. After careful consideration, we respectfully refrain from implementing a 'mechanistic diagram', primarily due to the risk of oversimplification and/or overinterpretation of our results. The reviewer raised this point in one of their earlier comments and we wholeheartedly agree: No single study can provide sufficient mechanistic insight to explain the complexities at the systems neuroscience level, and we fear that a necessarily oversimplified diagram would suggest exactly the opposite. We did aim to structure the 'story' behind our findings more clearly in our revision, both regarding their reporting in the main text and the figures and we explicitly welcome the reviewer's feedback in this regard.

Breath holds and respiratory phase mapping in circular statistics:

Please clarify how respiratory phase was derived and mapped. In many pipelines inhalation and exhalation are each assigned 180° , yet inspiratory and expiratory durations typically differ within and between participants, and brief breath holds are common. This asymmetry can bias phase-resolved effects if time spent in each phase is not represented accurately. Plus, State how holds and pauses were detected and handled (e.g., excluded, merged with exhalation, or treated as a distinct category). Were these even present?

Thank you for raising the important methodological issue of respiratory phase extraction. As described in the Methods section and in keeping with our previous work (Kluger & Gross, 2021; Kluger et al., 2021; Kluger et al., 2023; Kluger et al., 2024; Kluger et al., 2025; Saltafossi et al., 2025), respiration phase was extracted using a two-point interpolation approach, meaning that peaks and troughs were detected in the respiratory time series and phase was interpolated from peak to trough and trough to peak, respectively. Although, as noted by the reviewer, this results in an equal distribution of respiratory wave length between inspiration and expiration, this approach is perfectly capable of accounting for intraindividual variation in inspiratory/expiratory durations. In fact, we are currently preparing a tutorial paper on robust statistics for respiratory data in which we compare different methods for extracting respiratory phase and point out the advantages of two-point interpolation compared to the widely used Hilbert transform. To the reviewer's point, during respiratory phase extraction, the analyst has to decide between two imperfect scenarios: They can either divide phase evenly between inspiration and expiration despite knowing that - in units of time - expiration is longer than inspiration. This results in a slight mismatch between the time and phase domains, but keeps an even distribution of experimental events within both inspiration and expiration. Alternatively, the analyst can assign more than 180° to the (slightly longer) expiratory phase. This approach more accurately represents the time-phase relationship but does not allow an intuitive interpretation of respiratory phase as representing physiological processes (e.g. 180° no longer corresponds to the functionally meaningful change from active inspiration to the passive release of air from the lungs during expiration). Moreover, during the critical step of sorting events of interest into respiratory phase

bins, the latter option results in an imbalance between inspiration and expiration, with fewer trials being assigned to inspiratory phase bins (compared to expiratory ones). While this does more accurately represent the slightly uneven durations of both respiratory states, it compromises the interpretability of phase-locked neural or behavioural effects. Therefore, in our research, we have long decided on the former approach - knowing that it does not accurately represent time, but phase.

As for the reviewer's final comment regarding breath holds, all respiratory time series were visually inspected for breath holds and none were detected. This has now been added to the Methods section (p. 21):

To account for occasional, unusually high-amplitude breaths (e.g., sighs), segments that exceeded a normalised amplitude of $z = \pm 2.5$ were linearly interpolated (i.e., clipped). This way, such artefacts did not bias the peak detection algorithm we subsequently applied to identify time points of peak inspiration (peaks) and peak expiration (troughs; *findpeaks* function in Matlab with minimal peak prominence set to 1). **In addition, individual respiratory time series were visually inspected for noticeable breath holds, but none were detected.**

Overall, We offer a favourable evaluation. The study is careful, the methodology is innovative, and the TPJ-to-insula connectivity result adds a novel piece to the respiration–brain–perception puzzle. With the minor clarifications and additions above, the manuscript will provide a stronger bridge between near-threshold laboratory paradigms and broader active-sensing accounts of respiration, brain dynamics and vision.

Once again, the reviewer's positive assessment and productive comments are highly appreciated.

Reviewer #3 (Remarks to the Author):

1. What are the noteworthy results?

This study provides compelling evidence that respiration actively shapes human perception rather than serving as a passive, automatic background rhythm. Chalas et al. show that individuals fine-tune their breathing patterns to optimize sensory performance, aligning inhalation phases with moments of heightened cortical excitability and arousal preceding predictable stimulus onset. Corresponding shifts in alpha- and beta-band activity, measured with MEG, indicate that breathing modulates both visual processing and motor readiness to near-threshold visual stimuli. At the network level, directed connectivity from the temporoparietal junction to the insula reveals a key interoceptive pathway through which respiration influences perceptual context. Overall, the work positions breathing as a fundamental element of active sensing in humans and provides a framework for understanding - and ultimately treating - disorders of perception, attention, and interoception.

2. Will the work be of significance to the field and related fields? How does it compare to the established literature? If the work is not original, please provide relevant references.

It adds on the existent knowledge about respiratory-related perceptual modulation by more precisely specifying how respiration modulates neural excitability and arousal to aid performance

on a perceptual sensory task. Previous papers (e.g., Perl et al 2019) have shown similar effects but this work takes it a step further.

3. Does the work support the conclusions and claims, or is additional evidence needed?

It supports the conclusions and claims but see below.

Answers to the questions below is required.

4. Are there any flaws in the data analysis, interpretation and conclusions?

See below

Do these prohibit publication or require revision?

They do not prohibit publication but consideration. See below

5. Is the methodology sound? Does the work meet the expected standards in your field?

Yes

6. Is there enough detail provided in the methods for the work to be reproduced?

Yes

We thank the reviewer for their highly positive overall assessment of our manuscript.

Remarks to the authors related to points 3 and 4 above.

This study employs a variation of the classic Posner task, a well-established paradigm for probing visuospatial attention. This is a strong choice to demonstrate the role of respiration in perceptual processing, especially through the use of both spatial and temporal cues and well-controlled stimuli (Gabor patches). The fixed 1.6-s interval before target onset was intentionally designed to allow participants to align their respiratory phase with the upcoming stimulus. However, this design also introduces potential limitations.

If respiratory modulation were such a strong enhancer of perceptual sensitivity, one might expect participants—particularly those with better temporal alignment—to start adjusting their breathing even before cue presentation. Because the 1.6 s interval was constant, predictable, and relatively short in respiratory terms, an optimal strategy could have been to reset breathing as soon as the fixation cross appeared, ensuring that a favorable phase angle recurred 1.6 s later regardless of cue appearance. In this scenario, cues could serve to fine-tune performance rather than drive large respiratory phase shifts, which occur on a slower timescale.

Although many conclusions rely on within-subject analyses (e.g., trials with stronger phase alignment showing lower detection thresholds), this fixed-interval predictability remains a concern. A less predictable design—or inclusion of a control condition with jittered pre-target intervals (e.g., 1–6 s)—would have provided a stronger test of the hypothesis. Alternatively, an entrainment paradigm presenting streams of visual stimuli with variable inter-stimulus intervals and target probabilities (e.g., Lakatos et al., 2008) would be well suited to examining respiratory entrainment in an inherently rhythmic active-sensing context. While the use of temporal cues partly addresses this issue, the effect was modest ($t = 2.0$, $p = 0.047$) compared with the spatial cue benefit ($t = 4.28$, $p < 0.001$), suggesting that participants derived limited advantage from temporal information under the fixed 1.6 s schedule. The result on pre-target pupil modulation (fig 1h) also supports

the claim that respiration adjusts in the absence of cues (larger pupil changes in C-T- than C+T-). In addition, the lack of condition X respiration phase interaction on directed RMBO network connectivity (fig 4g) also supports this claim.

We truly appreciate the reviewer's in-depth comments regarding the implications and limitations of our experimental design. We fully agree with their suggestion that a variable prestimulus interval will provide key insights in the interaction between the use of spatial and temporal information, we would briefly like to elaborate why the task had been purposefully designed in its present form: Our main goal was to provide (much-needed) clear evidence in favour of or against respiratory adaptation in a task with variable levels of predictability, i.e. to answer the rather general question 'Does the amount of available spatial temporal information overall modulate the respiration-brain connection?'. To answer this question, we believe that the fixed-interval design was the best choice - if there indeed is respiratory adaptation in response to different levels of information, this provides the easiest context for participants to adjust their breathing. As the reviewer correctly points out, however, our pattern of results suggests that the two domains of information are not used equally, and that the behavioural consequences suggest a differential processing of spatial vs temporal information. This is why we not only replicated the study with lateralized auditory stimuli (data collection ongoing) to investigate how our findings generalise across sensory modalities, but we are also currently preparing a follow-up study with variable prestimulus time windows to better understand at which point of the task timeline respiratory adaptation is initiated. This includes blockwise within-subject manipulation of variable vs fixed intervals for both the pre-cue and pre-stimulus interval. We hope that these upcoming investigations will shed further light on how the respiration-brain-behaviour link dynamically interacts with changing sensory contexts.

All that being said, we acknowledge the reviewer's well-founded comments and have amended the manuscript with a Limitations section to reflect these considerations. The corresponding paragraph reads as follows (p. 19):

Methodologically, further variations of perceptual tasks could explore the present effects not only across sensory domains (e.g. in audition), but also with variable timings during the prestimulus fixation period for more precise readouts of individual respiratory alignment. Moreover, the benefit of catch trials (i.e. no stimuli presented) could be exploited to compute established signal detection theory measures like false alarm rate and decision criterion.

Regarding hit rates, figure 1b shows that C+T+ is higher than C-T- and that C-T+ is higher than C-T-. However, C+T- looks higher than C-T- and maybe even higher than C-T+ also. Are these effects significant?

Thank you for spotting this inconsistency: While (in line with the reviewer's comment) the figure caption correctly stated a higher hit rate for C+T- compared to C-T-, the significance marker was misplaced in the figure itself. The figure now accurately indicates higher hit rates for C+T+ and C+T- compared to C-T-. Since C-T- was used as the reference condition, our report was restricted to comparisons involving C-T-. The only other significant difference in hit rate was found for C+T+ vs C-T+, which has now been added for the sake of completeness.

Fig. 1. [...] **b**, Pairwise comparisons with Tukey-Cramer correction revealed significantly higher hit rates for C+T+ ($p < .001$) and C+T- ($p = .01$) compared to C-T-. In addition, we observed significantly higher hit rates for C+T+ compared to C+T- ($p = .01$). [...]

Fig 1g:

What are the units in the y-axis (pupil diameter)? Why they start between .2 and .3? If deflections downwards mean pupil constriction, then temporal cues (T+) constrict the pupil during the pre-target period up to target onset. This effect is even larger when temporal cues are presented alone (larger constriction in C-T+ compared to C+T+). Not sure this is clearly explained. Lighting was controlled across conditions discarding luminance changes. Were there fixation differences across conditions (e.g., more saccades to temporal cues in C-T+ than C+T+) or is the effect explained by arousal differences? Why C-T- shows more pre-target pupil constriction than C+T-?

Pupillary responses in Fig. 1g are provided in z-scores, as our pupil preprocessing included normalisation across the continuous time series prior to segmentation. Initial values reflected arbitrary units corresponding to the amount of pixels detected by the EyeLink camera within the pupil. In response to the reviewer's comment, Fig. 1 has been updated (kindly see our response to the previous comment), the figure legend has been amended, and a dedicated paragraph for the pupil preprocessing has been added as follows (p. 21):

Fig. 1. [...] g, Controlling for the presence or absence of temporal cues, spatial cues elicited robust increases in pupil diameter. Pupil diameter at target onset was strongly correlated with respiration phase (see inset; dashed line shows group-level mean across phase bins). Independently of temporal cues, spatially cued trials on average lead to significantly larger pupil diameter during the fixation period. **Note that pupil diameter is given in robust z-scores since pupil preprocessing involved normalization across the continuous time series. [...]**

Pupil preprocessing. Pupil data were preprocessed following procedures of previous work¹⁸. In short, area traces were converted to pupil diameter to linearize our measure of pupil size. Blinks were identified and visually validated by an automatic procedure (available from https://github.com/anne-urai/pupil_preprocessing_tutorial) and linearly interpolated. Blink-interpolated pupil time series were subjected to the procedure again using a relaxed criterion of $z = 6$ SD to capture remaining artifacts. Next, canonical responses to blinks were estimated and removed from pupil time series. To that end, pupil time series were band-pass filtered (pass band: 0.01–10 Hz, second-order Butterworth, forward-reverse two-pass). For subsequent analyses, pupil diameter time series were converted to z scores using a robust procedure (MATLAB function 'normalize' with options set to 'zscore' and 'robust').

Regarding the constriction effects following temporal cues, it is difficult to say with absolute certainty that lighting conditions did not change between e.g. C-T- and C-T+. If one were picky from a psychophysical standpoint, one aspect to note is that the temporal cue (i.e. the circular 'countdown' surrounding the central fixation marker) is an additional light stimulus appearing on a dark background. Pupillary constrictions for C-T+ (vs C-T-) and C+T+ (vs C+T-) may at least in part be explained by the additional brightness of the temporal cue, which would fit our pattern of results in Fig. 1g. This is why we refer to 'constant lighting conditions' in the strict sense only in the analyses we report, meaning contrasting conditions in which temporal cues are either present or absent in both cases. Following the reviewer's remarks, we have amended the manuscript to make this reasoning more clear (p. 6):

Controlling for lighting changes on the screen (i.e. the presence of the temporal countdown as a small, but light stimulus during C-T+ and C+T+ trials), spatial cues significantly modulated arousal during prestimulus fixation leading up to target presentation, irrespective of temporal cues (FDR-corrected $p < .05$; Fig. 1g), most likely reflecting the pre-stimulus difference in target uncertainty.

Finally, as for the reviewer's question regarding arousal or fixation differences, we do indeed interpret our results as a reflection of differential arousal states depending on target-related uncertainty and available spatiotemporal information. Since participants were not only instructed to fixate on the central marker, but we also had continuous real-time monitoring of both the pupil

ellipsoid (in the eyetracker) and the resulting time series (from the MEG), we have no reason to believe that saccade behaviour differed between conditions. As is routine in all our studies, participants are reminded of the instructions in case their ocular behaviour (fixations, blinks) systematically disrupts the experiment.

To confirm, we conducted an analysis of saccadic patterns across conditions. Raw saccade data from the eyetracker was segmented according to the trial definition of our behavioural analysis, cropped to the [-1.6 0] s prestimulus window, and sorted into different conditions. Using Fieldtrip's built-in artifact detection functionality *ft_artifact_zvalue*, we extracted the number of saccades exceeding a magnitude threshold of $z = 5$. Upon visual inspection, one participant was left out of this analysis due to an overly noisy signal in one of the experimental blocks. As expected, the number of saccades overall was low (with the group-level median ranging from 4-6 saccades over 180 trials across conditions, see plot below) and a Kruskal-Wallis test showed no group-level differences in the number of saccades across conditions ($p = .92$).

We have added this control analysis to the Results section as follows (p. 7):

As a control analysis, we quantified saccadic behaviour during the prestimulus time window across the four conditions. As expected, the number of saccades overall was low (with the group-level median ranging from $M = 4$ to $M = 6$ saccades over 180 trials across conditions) and a Kruskal-Wallis test showed no group-level differences in the number of saccades across conditions ($p = .92$).

Previous studies have shown that during spontaneous ventilation (no stimuli) pupil size expands during inhalation and contracts during exhalation. Namely it starts expanding 1-1.5s after inhalation onset. Are the present arousal-related pupil changes correlated to the subsequent inhalation-related pupil changes after target onset? Given that in this study, most subjects align their mid/late inhalations to target onset to improve detection, were there larger inhalation-related pupil changes in resp aligned trials?

Thank you for pointing out the intriguing connection between arousal-related pupil responses and respiration phase. As recently highlighted in a comprehensive review by Schaefer and colleagues (Eur J Neurosci 2022), the relationship between respiratory and pupillary time series is far from trivial with very mixed findings overall. In our data, we indeed report greater pupil diameter indicating higher arousal during inspiration, so the reviewer asks an important question regarding diameter change differences as a function of respiratory alignment. At this point one has to be careful not to conduct a circular analysis on the single-trial level, since we did not register post-stimulus inhalation for each trial (which might lead to a biased selection of trials to include in such an analysis).

In order to approximate the suggested analysis, we recomputed respiration phase-locked pupil diameter separately for each condition. To account for baseline differences in pupil diameter at target onset (see Fig. 1g), condition-specific vectors of pupil diameter ~ respiration phase were mean-centered within each condition. From Fig. 1c and 1e, it is evident that the C+T+ shows both the clearest behavioural effect (i.e. lowered inspiratory threshold) and the most focal distribution of trial onsets in that beneficial phase (i.e. the strongest respiratory adaptation). If there was a systematic link between preferred phase and inspiration-locked arousal changes, the increase in pupil diameter should be most prominent in C+T+ trials. This was not what we observed, however - the inspiratory increase in pupil diameter was highly consistent across conditions (see plot below).

The observation of condition-general inspiratory increase in arousal has been added to the manuscript as follows (p. 7):

Pupil diameter at target onset (**across all conditions**) was systematically related to respiration phase at target onset (Rayleigh test of individual circular means: $z(29) = 8.51$, $p < .001$), with greater diameter indicating higher arousal during inspiration. **In addition to the reported overall effect, this modulation was consistently observed within each condition as well.**

“Controlling for lighting changes on the screen, spatial cues significantly modulated arousal during prestimulus fixation leading up to target presentation, irrespective of temporal cues (FDR-corrected $p < .05$; Fig. 1g).” Does that mean that spatial cues prevent pupil constriction from happening during the pre-target period?

Thank you for prompting us to clarify that interpretation. We understand the pupil diameter pattern shown in Fig. 1g as follows: When a spatial cue is added, either from C-T- to C+T- or C-T+ to C+T+, we indeed observe a reduced constriction of the pupil, no matter whether a temporal cue is present or not. In other words, pupillary responses to spatial cues appear to reflect an 'upholding' of increased prestimulus arousal up to the point of stimulus presentation.

"Pupil diameter at target onset was systematically related to respiration phase at target onset (Rayleigh test of individual circular means: $z(29) = 8.51$, $p < .001$), with greater diameter indicating higher arousal during inspiration". Is the greater pupil constriction in C-T+ compared to C+T+ at pre-target and target onset related to higher arousal due to higher uncertainty about the target?

This is difficult to answer, and we were deliberately careful not to overinterpret these findings, but we would in principle agree with this interpretation. We clearly observe the higher arousal level indicated by greater pupil dilation, but can - strictly speaking - only speculate regarding the underlying mechanism. Since, as the reviewer points out, the uncertainty manipulation is the only experimental difference between the two conditions, we are confident that it would be difficult to come up with a convincing counterargument.

In response to the reviewer's remarks, we have added a brief (and still cautious) statement to clarify how we interpret this finding (p. 7):

Controlling for lighting changes on the screen, spatial cues significantly modulated arousal during prestimulus fixation leading up to target presentation, irrespective of temporal cues (FDR-corrected $p < .05$; Fig. 1g), **most likely reflecting the pre-stimulus difference in target uncertainty.**

Using a mutual information model, the authors showed that near-threshold visual detection can be predicted not only by spatial-temporal cues, but also by the respiratory phase at stimulus onset. Incorporating the sine and cosine of the peristimulus breathing cycle significantly improved single-trial prediction accuracy, demonstrating that both stimulus predictability and ongoing respiratory state jointly shape perceptual outcomes. This analysis robustly replicates and extends prior reports of respiration-linked perceptual modulation.

It is hard to grasp some of the aspects in Fig 1f. Black dots seem quite widely distributed in the y-axis. Does this mean that preferred resp phase is different in each subject? This interpretation contrast with fig 1c – in which mid inhalation is the ideal timing across all subjects (e.g., lower contrast gabor patches are perceived if presented during mid inhalation) and fig 1e. Maybe showing the raw or averaged respiration waveforms overlaid would be helpful.

We apologize for the perceived inconsistency between Fig. 1f and results shown in panels 1c+e. From our description, it was apparently not always clear which conditions we referred to. To reiterate, Fig. 1c shows the C+T+ condition driving the inspiratory facilitation effect, which is highly consistent with Fig. 1e (where these trials are contrasted against C-T- trials). The grey arrow pointing from Fig. 1e to 1f was admittedly distracting, as panel 1f shows different conditions (the arrow has been removed accordingly): Dark blue dots show the overall relationship between

preferred phase and respiration rate changes **for all conditions combined**, which is why the distribution of preferred phase appears more scattered and cannot be compared to the one shown in panel 1e.

To remedy this potential misunderstanding of our analyses, we have adjusted both Fig. 1 as well as the main text. While the updated Fig. 1 with the grey arrow removed and clearer axis labels can be seen above, the respective passages in the main text read as follows:

Fig. 1. [...] **f**, Group-level correlation of respiratory adaptation with task performance. The more strongly participants adapted their respiratory frequency during the task (compared to rest), the better their performance **for timed trials (light blue)**. A strong circular-linear correlation between rest-to-state changes in respiration rate and preferred phase **across all conditions** suggests respiratory adaptation in favour of behavioural facilitation (blue). [...]

Moreover, **across all conditions**, the more strongly participants adapted their respiratory frequency during the task (compared to rest), the better their performance (i.e. the lower the contrast necessary to achieve the targeted hit rate; see Fig. 1f). Critically, circular-linear correlation analysis revealed a robust positive correlation between rest-to-task change in respiration rate and preferred phase at stimulus onset for **timed-only stimuli (C-T+; ρ (28) = .50, $p = .023$)**. (p. 6)

The result that respiratory rate during task is higher than rest could be linked to the fact that it is easier to make phase adjustments in short resp cycles compared to longer ones.

Regarding the analyses of the neural data, I believe the authors used state-of-the-art analyses and the results are strong with impressive correlations between neural activity in the RMBO network, performance and respiratory phase.

Thank you for this suggestion. Faster respiratory rate during tasks compared to rest is a finding that has been replicated many times. The reviewer is entirely correct that one explanation could be a 'baseline readiness' due to shorter overall cycles in order to even more flexibly adjust one's respiratory cycles to external task demands (which are absent during rest). Our finding of greater differences from rest to task being predictive of perceptual performance support such an interpretation.

Given the activation of temporal site (TPJ)– did the participants heard their breathing sounds modulating auditory cortex activity? did they wear headphones?

Participants were seated in a quiet environment inside the MEG chamber and were therefore not wearing headphones. Note that the source-localized activity within TPJ on the cortical surface (parcels # 61 & 167 in the HCP atlas) is located rather far away from primary or associative auditory cortices below the temporal lobe (parcels # 37 & 152).

It is known that hyperventilation can modulate neural excitability. Given that participants breathed faster during the task compared to rest, could this have influence some of their results? Was tidal volume measured or controlled in anyway?

Tidal volume was constantly monitored with a respiration belt and participants were monitored via video throughout the MEG recording. No participants showed any signs of hyperventilation.

Reviewer #4 (Remarks to the Author):

We thank both the ECR reviewer as well as their mentor for their work in evaluating our manuscript and appreciate the journal's approach in strengthening junior researchers' contributions to the scientific process.

Response to Reviewers

We would like to thank all reviewers for their positive feedback and for supporting publication of our manuscript. In what follows, we address the final minor points raised by Rev 2. Reviewer comments are shown in **grey**, our responses in **blue**, and quoted changes to the manuscript in **red**.

Reviewer #2:

I would first like to thank the authors for their careful and thoughtful responses to the previous round of comments. The revised manuscript is substantially improved. The following points are offered as minor comments aimed at further strengthening clarity, scope, and interpretative precision. None of them detract from the overall quality or validity of the work.

Thank you, we appreciate the reviewer's positive encouraging assessment of our revision.

Framing of active sensing and voluntary respiratory control: While the evidence for task-dependent respiratory adjustment and respiration-locked behavioural facilitation is compelling, some passages still read as slightly stronger than strictly warranted in terms of voluntary control. The data clearly support adaptive coupling between respiration, neural dynamics, and perceptual performance, but direct causal evidence for strategic respiratory control remains inferential.

It may therefore be helpful to slightly temper the language in places, e.g., by emphasizing adaptive alignment or task-dependent adjustment rather than fully intentional control.

We agree that even a compelling body of evidence should not be overinterpreted as a clear indication of voluntary control (excluding any other potential explanation). In line with the reviewer's suggestion, we have therefore rephrased the manuscript in the following instances to emphasize alignment/adjustment:

Participants adapted their breathing patterns to align with predictable stimulus onset, and this adaptive respiratory alignment correlated with improved performance. (p. 1)

Together with the behavioural evidence outlined above, the tight link of these respiration-related excitability modulations to sensory sampling as well as their flexible adaptation to predictability changes strongly suggest that the respiratory rhythm is adaptively aligned with task dynamics in order to form an increasingly precise predictive model of upcoming sensory information and optimally prepare corresponding behavioural responses. (p. 16)

Behavioural interpretation and perceptual sensitivity: Given the absence of catch trials, established signal detection theory measures such as false alarm rate and decision criterion cannot be computed. While this limitation is acknowledged later in the manuscript, it may be useful to briefly remind the reader earlier that respiration-related threshold effects could, in principle, reflect a combination of sensory sensitivity and response criterion shifts.

Thank you, we have now added an earlier statement regarding this limitation:

It should be noted that phase-related changes in decision criterion could occur simultaneously with the sensitivity modulation, but their computation would require changes in the experimental paradigm (i.e. the introduction of catch trials). (p. 4)

Related literature and completeness: For the sake of completeness, the authors may wish to consider citing a recent preprint that is closely related in scope and conceptual framing: https://osf.io/preprints/psyarxiv/zj23s_v1

We appreciate the suggestion and have gladly added the new preprint as reference #30 to the Introduction section:

At present, converging evidence from different lines of research strongly suggests respiratory involvement in active sensing, including adaptation of respiratory behaviour to task timing ²⁸, modulation of excitability and arousal states ⁴, and behavioural facilitation ²⁹⁻³⁰ (p. 3)

Interpretation of directed connectivity analyses: The multivariate Granger causality analyses are carefully implemented and constitute a clear strength of the manuscript. That said, as with any GC-based approach, directed connectivity estimates may still be influenced by source leakage, common drivers, and slow rhythmic modulation.

A brief, explicit note acknowledging these general limitations, and clarifying that the reported directionality reflects statistical predictability rather than anatomical or causal direction, would further strengthen interpretative caution and transparency. This would be particularly useful in sections linking specific INS–SMA or TPJ–V1 interactions to functional roles.

The question of causality closely relates to a previous comment regarding voluntary control. In keeping with our previous response, we have added a cautionary statement with regard to the limits of GC analyses and their interpretation in the Discussion section:

Note that any directional connectivity analysis cannot strictly exclude influences like source leakage or infraslow modulation. Hence, even the compelling phase-related directionality changes presented here reflect statistical predictability rather than immediate anatomical or causal direction. (p. 18)

Density and presentation: The manuscript is necessarily rich, but some sections, particularly in the Results and Discussion, remain quite dense and occasionally reiterate similar points across behavioural, oscillatory, and connectivity levels. A modest reduction of repetition or slightly clearer "marking" of which analyses are confirmatory versus novel could improve readability without loss of content.

In response to the reviewer's suggestion, we have rephrased the manuscript in a few instances in order to more clearly separate novel insights from corresponding (but probably confirmatory) results in other modalities:

Confirming the performance effects shown in Fig. 1b, including spatial and temporal cues significantly improved the binary prediction of correct vs incorrect responses compared to a base model that only used single-trial contrast information ($t(29) = 2.64$, $p = .013$). (p. 4)

Confirming the initial behavioural results, we did in fact observe clustering of single trials during the inspiratory phase for highly predictable C+T+ trials (Hodges-Ajne test against uniformity: $p = .005$), but not for C-T- trials ($p = .560$). (p. 6)

Moving from arousal neuromodulation to excitability states, we next sought to confirm a well-established neural signature of perceptual performance, namely alpha suppression effects over parieto-occipital sensors for hits vs misses (i.e., perceived vs non-perceived trials; Fig. 2a). Replicating previous work from our lab 4, we report long-lasting alpha suppression for perceived trials (Fig. 2b), starting as early as the cue onset at -1600 ms prestimulus. (p. 7)

First, we sought to confirm the overall effect of respiratory phase on prestimulus spectral power within our network of interest. (p. 8)

In summary, this is a strong and carefully executed study that makes a meaningful contribution to our understanding of respiration as a modulator of perception and neural dynamics. The points above are intended solely as minor refinements. I commend the authors for the quality of the work and for the clear improvements introduced in the revised version.

Thank you again for consistently helpful and productive comments, much appreciated.